# Spatial transcriptomics analysis of esophageal squamous precancerous lesions and their progression to esophageal cancer

Xuejiao Liu[1,2,7], Simin Zhao[2,3,4,7], Keke Wang[2,7], Liting Zhou[1,2,4], Ming Jiang[2], Yunfeng Gao[2], Ran Yang[2], Shiwen Yan[1,2,4], Wen Zhang[1,2,4], Bingbing Lu[1,2,4], Feifei Liu[1,2,4], Ran Zhao[1,2], Wenting Liu[1,2], Zihan Zhang[1,2,4], Kangdong Liu[1,2,4,5,6], Xiang Li ®[1,2,4,5,6] ✉ & Zigang Dong ®[1,2,4,5,6] ✉

Esophageal squamous precancerous lesions (ESPL) are the precursors of esophageal squamous cell carcinoma (ESCC) including low-grade and high-grade intraepithelial neoplasia. Due to the absence of molecular indicators, which ESPL will eventually develop into ESCC and thus should be treated is not well defined. Indicators, for predicting risks of ESCC at ESPL stages, are an urgent need. We perform spatial whole-transcriptome atlas analysis, which can eliminate other tissue interference by sequencing the specific ESPL regions. In this study, the expression of TAGLN2 significantly increases, while CRNN expression level decreases along the progression of ESCC. Additionally, TAGLN2 protein level significantly increases in paired after-progression tissues compared with before-progression samples, while CRNN expression decreases. Functional studies suggest that *TAGLN2* promotes ESCC progression, while *CRNN* inhibits it by regulating cell proliferation. Taken together, *TAGLN2* and *CRNN* are suggested as candidate indicators for the risk of ESCC at ESPL stages.

Esophageal cancer is one of the most aggressive and lethal malignant tumors in the world. It is mainly classified into two subtypes: esophageal squamous cell carcinoma (ESCC) and esophageal adenocarcinoma (EAC)[1-3]. ESCC is mainly distributed in East Asia and is the major type of esophageal cancer occurred in China[4]. Esophageal squamous precancerous lesions (ESPL) are considered potential precursors of ESCC, exhibiting a progression from normal epithelia (NE) to low-grade intraepithelial neoplasia (LGIN), high-grade intraepithelial neoplasia (HGIN), and eventually developing into invasive cancer. According to the extent of the cancer invasion into the esophageal mucosa, LGIN can be divided into mild dysplasia and

moderate dysplasia, and HGIN including severe dysplasia and carcinoma in situ[5,6].

In a cohort of 682 patients who underwent endoscopy, 114 individuals (16.7%) were diagnosed with ESCC during a 13.5-year follow-up period. The relative risks (with 95% confidence intervals) for developing ESCC, based on the initial histological diagnosis, were as follows: normal, 1.0 (reference); mild dysplasia, 2.9 (1.6–5.2); moderate dysplasia, 9.8 (5.3–18.3); severe dysplasia, 28.3 (15.3–52.3); and carcinoma in situ, 34.4 (16.6–71.4)[5]. Notably, a clear trend was observed, with higher grades of dysplasia associated with a significantly increased risk of developing ESCC. Therefore, studying the pathogenesis from ESPL

---

[1]Department of Pathophysiology, School of Basic Medical Sciences, Zhengzhou University, Zhengzhou, Henan, China. [2]China-US (Henan) Hormel Cancer Institute, Zhengzhou, Henan, China. [3]Department of Pathology, Affiliated Cancer Hospital of Zhengzhou University, Zhengzhou, Henan, China. [4]Tianjian Laboratory of Advanced Biomedical Sciences, Institute of Advanced Biomedical Sciences, Zhengzhou University, Zhengzhou, Henan, China. [5]The Collaborative Innovation Center of Henan Province for Cancer Chemoprevention, Zhengzhou, Henan, China. [6]State Key Laboratory of Esophageal Cancer Prevention and Treatment, Zhengzhou University, Zhengzhou, Henan, China. [7]These authors contributed equally: Xuejiao Liu, Simin Zhao, Keke Wang. ✉e-mail: lixiang@zzu.edu.cn; dongzg@zzu.edu.cn

to ESCC and understanding its underlying mechanisms are helpful to clarify the etiology of ESCC development and will provide an important foundation for early prevention strategy discovery.

Significant improvements in diagnostic technology, particularly early detection, have contributed to the reduction in cancer mortality rates. Nevertheless, there is currently a lack of precise indicators for the early diagnosis of ESCC. Presently, the conventional approach to investigate ESCC indicators primarily involves the analysis of differentially expressed genes (DEGs) between normal and esophageal cancer tissues[7]. Although these indicators can be valuable in predicting prognostic outcomes like overall survival and disease-free survival, they may not be suitable as indicators for ESPL. Moreover, they may not adequately demonstrate the potential for ESCC development in patients initially diagnosed with LGIN or HGIN.

Currently, publications have only reported the genomics[8] and DNA methylation of ESPL[9,10]. There are few reports on the pathogenesis from ESPL to ESCC, primarily due to the following reasons: (1) Fresh ESPL tissues, typically obtained from biopsies, are not sufficiently large to support multi-omics analysis; (2) The majority of the tissue is composed of normal epithelia (NE), with only a small region of the ESPL tissue, especially at the early stage of precancerous lesions, representing the actual ESPL. When using these tissues for bulk RNA-seq, the data may be imprecise. Therefore, advanced technology is an urgent need to study the progression of ESCC. Spatial transcriptomics, which preserves spatial location, offers vital information for studying the relationships between cell function, phenotype, and microenvironment[11,12]. Utilizing the spatial transcriptome sequencing method will enable the selection and sequencing of the authentic ESPL regions without interference from other tissues.

Clinically, LGIN patients usually need to inspect once a year or several years later without surgical treatment. HGIN patients are suggested to do endoscopic submucosal dissection or endoscopic mucosal resection. Considering the incidence of esophageal cancer in individuals with a normal esophagus and LGIN, it is impractical to leave all LGIN patients untreated. Thus, indicators, that can predict risks of ESCC when people are diagnosed with ESPL, as well as distinguish whether the LGIN patient needs treatment or not to prevent progression of ESCC, are necessary. For this reason, we select the real ESPL and invasive cancer parts as regions of interest for spatial whole-transcriptome atlas (WTA) analysis from paraffin-embedded tissue sections to find genes that accelerate ESPL transition to ESCC.

In this study, ESPL status is analyzed. Immune microenvironment analysis reveals an immune suppressive condition in ESPL stages. Principal component and DEGs analysis indicate that cancer-like changes primarily initiate at the HGIN stage. Certain DEGs demonstrate progressive increase or decrease throughout the disease progression from ESPL to ESCC. Machine learning methods are employed to shortlist potential candidate genes. These genes are confirmed through IHC staining, revealing a significant increase in TAGLN2 expression and a decrease in CRNN expression across pathological stages. Validation using paired before-progression and after-progression tissue confirms the correlation of TAGLN2 and CRNN with ESCC progression. Cell proliferation analysis, colonies formation assay, and in vivo study illustrate the functional role of these candidate genes in the progression of ESCC. Based on our study, early intervention is recommended upon detecting aberrant expression of TAGLN2 and CRNN, even at the LGIN stage. Our findings could contribute to the prevention of esophageal cancer.

## Results
### Spatial whole-transcriptome profiling of ESPL
To characterize the pathogenesis from ESPL to ESCC, we conducted spatial whole-transcriptome profiling, as depicted in Fig. 1a. The entire process involves probe hybridization with photocleavable oligo-conjugated antibodies, selection of regions of interest (ROIs), photocleavage, oligos collection, and gene expression sequencing. Serial sections of tissue samples were used for H & E staining and WTA analysis. Figure 1b illustrates a schematic diagram of ESCC progression. In most cases, dysplasia covering over half of the esophageal mucosa is classified as HGIN, otherwise, it is considered LGIN. ESCC is characterized as a type of cancer where cancerous cells breach the basal layer. Representative WTA-analyzed tissue samples with H & E staining or fluorescent labeling are shown in Fig. 1c. ROIs were selected based on H & E and Pan-cytokeratin staining. Regions of NE, LGIN, HGIN, and ESCC are shown in the appointed area. The region with the maximum ROI was selected to obtain the largest lesion area for sequencing. WTA sequencing was conducted with these tissues from selected ROIs. In this WTA experiment, all the ROIs passed technical signal quality control and technical background quality control (Supplementary Fig. 1a–d).

### ESPL status analysis
Investigating the status of these precancerous lesions is essential for understanding the etiology and mechanisms of ESCC progression. This study focused on analyzing the status of ESPL as well as ESCC stage from several aspects, including biological processes that occur during tumorigenesis, regulatory network, abnormal metabolic pathways that occur in tumors, abnormal signal transduction pathways in cancer, and immune-related pathways (Fig. 2a). Cancer development involves a complex interplay of cellular processes. Among these processes, mitotic chromosome condensation[13], mitotic sister chromatid cohesion, and DNA conformation change[14] are vital for ensuring proper chromosome segregation during cell division. In normal cells, these processes are tightly regulated, but in cancer cells, they can become dysregulated, leading to abnormal chromosome segregation and genomic instability. The increase in mitotic chromosome condensation may promote the development of genetic mutations or chromosomal abnormalities commonly observed in cancer cells. Additionally, increased mitotic sister chromatid cohesion can lead to improper chromosome alignment, resulting in aneuploidy, which is a hallmark of many types of cancer. Finally, increased DNA conformation change can alter gene expression patterns, contributing to the development of cancer. Therefore, an increase in these processes during tumorigenesis stages suggests that they may play important roles in the transformation from ESPL to ESCC.

Cell cycle and the p53 signaling pathway are integral processes in cancer development[15]. In cancer cells, cell cycle can become dysregulated, resulting in uncontrolled cell growth and division. Similarly, the p53 signaling pathway is one of the body's primary defense mechanisms against cancer. This pathway is activated when DNA damage is detected, allowing the cell to repair the damage or undergo apoptosis. Unfortunately, in cancer cells, the p53 pathway can become dysregulated, allowing damaged cells to continue growing and dividing uncontrollably. Overall, the dysregulation of these processes leads to the accumulation of genetic mutations or chromosomal abnormalities, contributing to uncontrolled cell growth and division. The enrichment of these two pathways increases with ESCC progression, which is consistent with the pattern of ESPL progression to ESCC.

Several abnormal metabolic processes[16] increased during ESCC progression including purine, pyrimidine metabolism, and citric acid cycle. The pathway of nucleotide metabolism is important as it is responsible for the production of purine and pyrimidine molecules that are necessary for tasks such as DNA replication, RNA synthesis, and cellular bioenergetics. When the rate of nucleotide metabolism is increased, it can lead to uncontrolled growth and proliferation of tumors, which is a common characteristic of cancer. As such, elevated nucleotide metabolism is often regarded as a hallmark sign of cancer. The citric acid cycle is essential for generating cellular energy by breaking down various nutrients, including glucose, amino acids, and fatty acids. This cycle can also support the Warburg effect[17] by

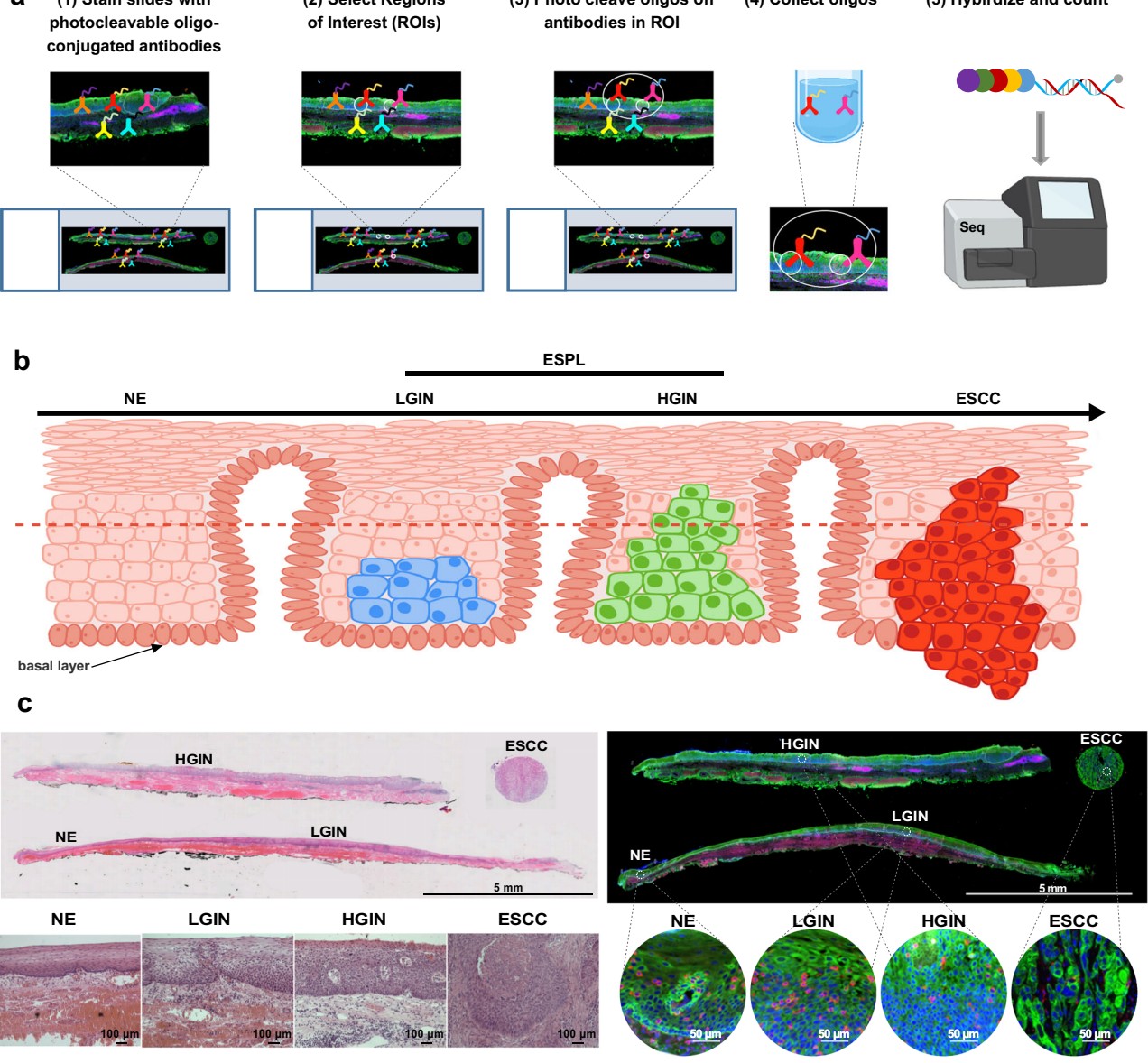

**Fig. 1 | Spatial whole-transcriptome profiling of ESPL. a** Schematic of the WTA analysis. The whole process includes probe hybridization with photocleavable oligo-conjugated antibodies, ROIs selection, photocleavage, oligos collection, and gene expression sequencing. **b** Schematic diagram of ESCC progression. Dysplasia covering over half of the esophageal mucosa is classified as HGIN, otherwise, it is considered LGIN. ESCC is recognized as the cancer cell breaking through the basal layer. **c** Representative tissue samples with H & E staining (Scale bar, 5 mm, and 100 μm, representative of $n = 3$ independent experiments), or fluorescent labeling for WTA analysis (Scale bar, 5 mm and 50 μm). Morphologically NE, LGIN, HGIN, and ESCC regions as well as partially magnified images of ROIs are appointed in the picture. Pan-cytokeratin (green), CD45 (pink), and Syto13 (blue) are stained for epithelial cells, immune cells, and nucleus, respectively.

providing intermediates required for biosynthesis pathways like lipid and nucleotide synthesis. As a result, this process can facilitate cancer growth.

Cancer-related abnormal signal transduction pathways including EGFR, FGFR, WNT, TGF-beta, and MAPK are activated during ESCC progression stages. Immune cells are an essential part of tumor initiation and development[18–20]. Tumor-associated fibroblasts (CAFs), Matrix and Matrix remodeling-related genes enriched from NE to ESCC. Matrix remodeling plays an important role in immune microenvironment reconstruction and CAFs greatly affect Matrix remodeling. The progression of tumors is strongly influenced by these factors[21].

We also analyzed the immune infiltration of ESPL and ESCC samples to explore the changes in the immune microenvironment by the Spatialdecon (Version 1.8.0) safeTME deconvolution method[22].

This method enables the quantification of immune cell types using spatial transcriptomic data. The heatmap of immune cell abundance in different ROIs is shown in Fig. 2b. Immune cell proportion across the four pathological stages was investigated to reveal the changes of immune cells during ESCC tumorigenesis (Fig. 2c–e). The data revealed an increase in fibroblasts and macrophages during the transition from ESPL to ESCC. Fibroblasts, a major component in the tumor microenvironment, are involved in the regulation of cancer cell proliferation and invasion as well as participating in immune regulation[23,24]. Macrophages play an important role in accelerating tumor progression[25]. Additionally, we observed a decrease in M1 macrophages, which have an anti-cancer effect, during ESPL stages, while M2 macrophages with cancer-promoting function increased, as analyzed by Cibersort (Version 1.04). To validate the expression of fibroblasts and macrophages in NE, LGIN, HGIN, and ESCC stages, we utilized specific markers

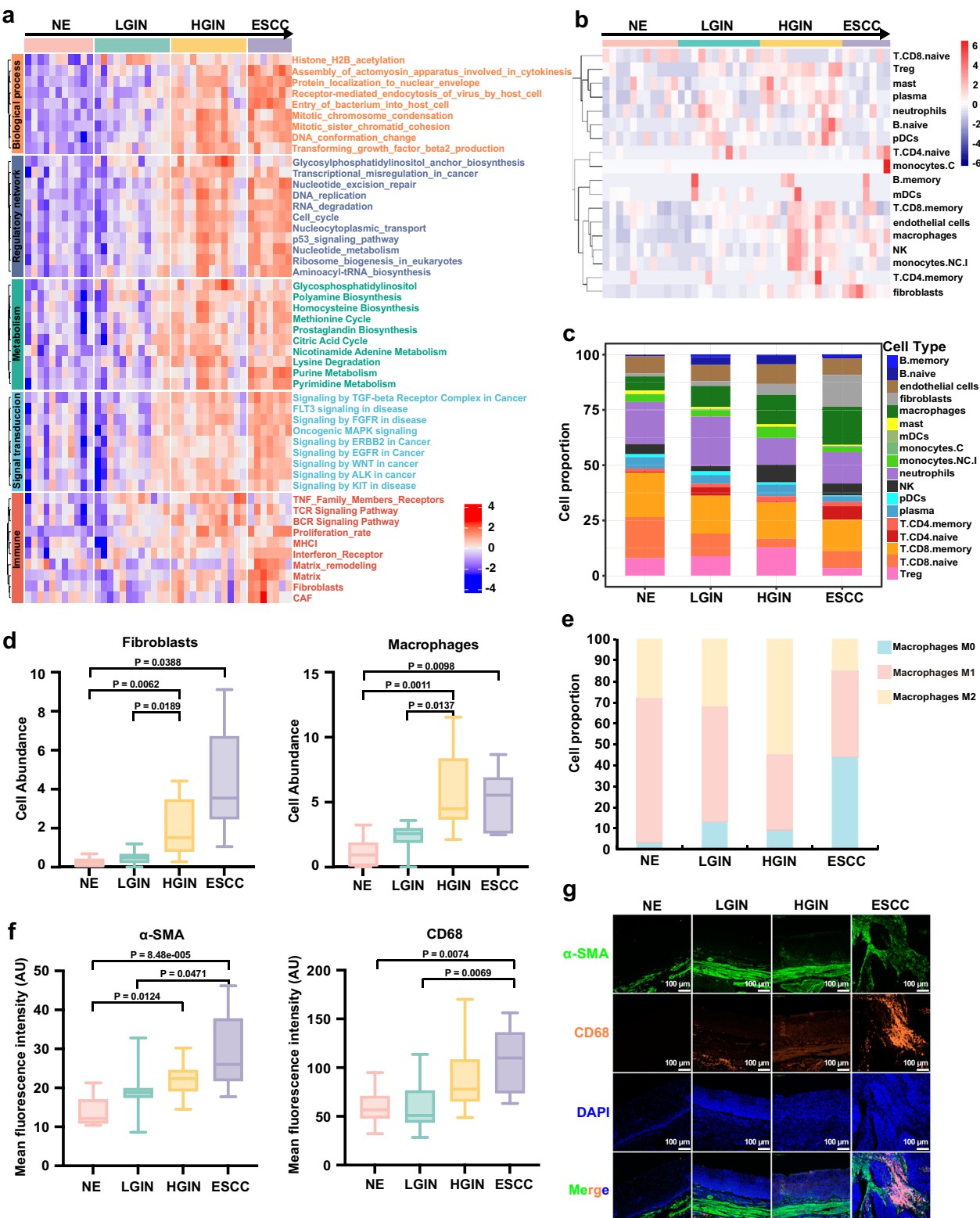

**Fig. 2 | Status analysis of ESPL. a** Heatmap of ESPL and ESCC in biological processes that occur during tumorigenesis, regulatory network, abnormal metabolic pathways that occur in tumors, abnormal signal transduction pathways in cancer, and immune-related pathways. **b** Immune cells abundance in each ROI. **c** Stacked histogram showing immune cell proportion across the four pathological stages. **d** Fibroblasts and macrophages cell abundance in precancerous and ESCC stages. Brown-Forsythe and Welch ANOVA test with Dunnett's T3 multiple comparisons for the comparison among groups. ROIs number: NE (n = 11), LGIN (n = 12), HGIN (n = 12), ESCC (n = 7). **e** Macrophage cell proportion is divided into M0, M1, and M2 macrophage cells.

**f** Validation of fibroblast and macrophage expression was performed by immuno-fluorescence using specific markers (α-SMA for fibroblasts and CD68 for macrophages). Sample size: NE (n = 10), LGIN (n = 10), HGIN (n = 10), ESCC (n = 10) biologically independent samples. Kruskal–Wallis test and corrected by Dunn's test for multiple comparisons. **g** Representative images of immunofluorescence staining (Scale bar, 100 μm). In the box plots (**d**, **f**), the boxplot shows the median (central line), upper and lower quartiles (box limits), and min to max range (whiskers). Source data are provided as a Source Data file.

(α-SMA for fibroblasts and CD68 for macrophages) in immuno-fluorescence staining. Our results demonstrated increased expression levels of α-SMA and CD68 across various disease stages, ranging from NE to ESCC (Fig. 2f, g), indicating an escalation in fibroblasts and macrophages abundance. Taken together, the results of fibroblasts, macrophages, and the increasing proportion of T regulatory cells (Treg) in LGIN and HGIN stages indicate an immune suppressive microenvironment during ESPL stages. In the ESCC stage, there is a decrease in the proportion of M2 macrophage cells and Treg cells compared to the ESPL stages, possibly due to the activation of the immune system in response to early-stage tumorigenesis as it actively combats the tumor. Our results illustrate the immunosuppressive condition in ESPL stages, potentially attenuating the anti-tumor immune response and accelerating ESCC tumorigenesis.

## Differentially expressed genes analysis from spatially defined regions of ESPL

Principal component analysis (PCA) by analyzing the spatially defined regions of ESPL showed a different expression profile among NE, LGIN, HGIN, and ESCC (Fig. 3a). The figure demonstrates clear separation between NE and ESCC samples. Furthermore, LGIN exhibited greater similarity to NE than HGIN, while HGIN displayed more similarity to ESCC than LGIN. These findings support the idea that HGIN carries a higher risk of progressing into ESCC compared to LGIN. The PCA analysis demonstrated that WTA analysis of spatially defined ESPL closely corresponds to the histopathological pheno-types of malignant progression stages during ESCC development. The presence of some LGIN samples in close proximity to HGIN samples suggests that certain LGIN tissues share a similar transcriptional background with HGIN, potentially indicating the possibility of progression from the LGIN stage to the HGIN stage, and ultimately ESCC. Hence, LGIN patients with a high risk of ESCC may require early treatment.

To identify DEGs that may imply the risk of ESCC when patients are diagnosed with LGIN or HGIN, we compared the DEGs among LGIN, HGIN, and ESCC groups. The gene expression heatmap and cluster analysis for each group are shown in Supplementary Fig. 2a, b. The upset plot revealed 21 up-regulated co-DEGs and 23 down-regulated co-DEGs that overlapped DEGs between LGIN vs NE, HGIN vs NE, and ESCC vs NE (Fig. 3b, c). The LGIN group exhibited a much smaller number of DEGs (only 63 genes) compared to the HGIN (826 genes) or ESCC group (1298 genes), suggesting fewer cancer-like characteristics and some similarities with NE features. This finding is consistent with PCA data.

The analysis of DEGs between different groups is shown in Fig. 3d. In the comparison between the LGIN group and NE group, significantly higher levels of *MAL*, *TMPRSS11B*, *EMP1*, *EPS8L1*, *ECM1*, and *KRT78* were observed in normal tissue, while the LGIN group exhibited higher expression level of *S100A7*, *KRT1*, *KRT16*, *KRT10*, *GSTA1*, and *RPN2*. Comparing the HGIN group with the NE group revealed up-regulated DEGs such as *KRT17*, *IGHG2/3/4*, and *IGKC*, whereas down-regulated DEGs included *SPRR3*, *KRT4*, *MAL*, *CRNN*, *CNFN*, and *TGM3*. The ESCC group exhibited high levels of *KRT10*, *KRT16*, *KRT17*, *KRTDAP*, and *IF16*, while showing low levels of *CRNN*, *MAL*, *KRT4*, *KRT13*, *KRT78*, and *SPINK5* when compared with the NE group. The gene expression heatmap of progressively changed co-DEGs among the LGIN, HGIN, and ESCC groups is presented, showing the candidate gene expression level in each group separately (Fig. 3e).

To elucidate the mechanisms in esophageal cancer pathogenesis, we performed pathway enrichment analysis among the LGIN, HGIN, and ESCC groups. Figure 3f displays the changes in signaling pathways between each two groups, while Supplementary Fig. 2c–e and Supplementary Fig. 3a-f show the pathway-related genes. The LGIN group exhibited involvement in the formation of the cornified envelope and keratinization signaling pathways compared with the

NE group. However, as the disease progressed to HGIN and ESCC stages, the enrichment of these signaling pathways-related genes decreased. In the HGIN group, the DEGs were found to be involved in signaling pathways such as DNA methylation, ERCC6 (CSB), and EHMT2 (G9a), which positively regulate rRNA expression and RNA polymerase I promoter opening, among others. The ESCC group showed similar changes in signaling pathways as the HGIN group, with differences mainly observed in the cornified envelope, keratinization, Ub-specific processing proteases, extracellular matrix organization, and syndecan interaction signaling pathways. The enrichment in the extracellular matrix organization signaling pathway aligns with our immune microenvironment analysis. Figure 4a presents rank-based gene set enrichment analysis (GSEA) of gradient-changed signaling pathways during ESCC tumorigenesis and representative genes. Our data suggest that gene expression and biological functions begin to alter during ESCC tumorigenesis at the LGIN stage. However, despite these changes, PCA and DEGs of LGIN show similarities to NE, while significant cancer-like changes primarily commence from the HGIN stage. Considering the possibility of progression from the LGIN stage to the HGIN stage, and eventually ESCC, early intervention for LGIN patients at high risk of ESCC is warranted.

For a comprehensive understanding of the esophageal epithelial cell status transitions and their correlation with DEGs, we employed pseudotime analysis[26,27] using SCORPIUS[28,29] (version 1.0.8) to simulate developmental trajectories of ESCC. The pseudotime analysis demonstrated a remarkable correspondence between esophageal epithelial cell transitions and the development stage of ESCC (Fig. 4b). We further enriched the heatmap of gradient-developed DEGs according to pseudotime and ESCC progression (Fig. 4c), along with the expression of representative genes correlated with pseudotime (Fig. 4d). Among our data, the formation of the cornified envelope and keratinization emerged as the most significantly changed signaling pathways, indicating their pivotal role during ESCC progression. As a result, we defined the Differentiation & Keratinization (D & K) score, utilizing a gene panel correlated with cell differentiation and keratinization, to predict the prognosis of ESCC (Supplementary Table 1). Notably, a low D & K score was associated with a poor prognosis. Furthermore, based on co-DEGs that progressively increase in LGIN, HGIN, and ESCC, we defined the Cancerization score (Supplementary Table 1). A high cancerization score was found to be indicative of a shorter survival time (Fig. 4e). All the genes utilized in the D & K score and Cancerization score analysis are selected from DEGs with the |log2(FC)| > 1 in ESPL stages, allowing for the prediction of patient prognosis when diagnosed with ESPL.

## Candidate indicators screening for predicting the risk of ESCC

The DEGs that exhibit progressive changes starting from the LGIN stage and continuing through all stages may potentially accelerate ESCC tumorigenesis. To identify potential candidate genes serving as risk indicators for the progression of ESPL to ESCC, we analyzed the expression levels of co-DEGs at NE, LGIN, HGIN, and ESCC stages. The co-DEGs that show progressive changes were considered as potential candidates. In order to differentiate between NE, LGIN, HGIN, and ESCC and identify genetic markers that can guide patient management and personalized treatment options for individuals at risk of developing ESCC from ESPL, it was crucial to establish the genes' ability to distinguish between these stages. Considering that PCA analysis indicated greater similarity between HGIN and ESCC and that patients with HGIN and ESCC require treatment, while those with normal mucosa and LGIN do not, we divided our WTA sequenced samples into three distinct groups. These groups consisted of a normal group (N), a low-grade group (L) including LGIN, and a malignant group (M) comprising both HGIN and ESCC.

K-nearest neighbor (knn), logistic regression (logre), support vector machine (svm), random forest (rf), neural network regression

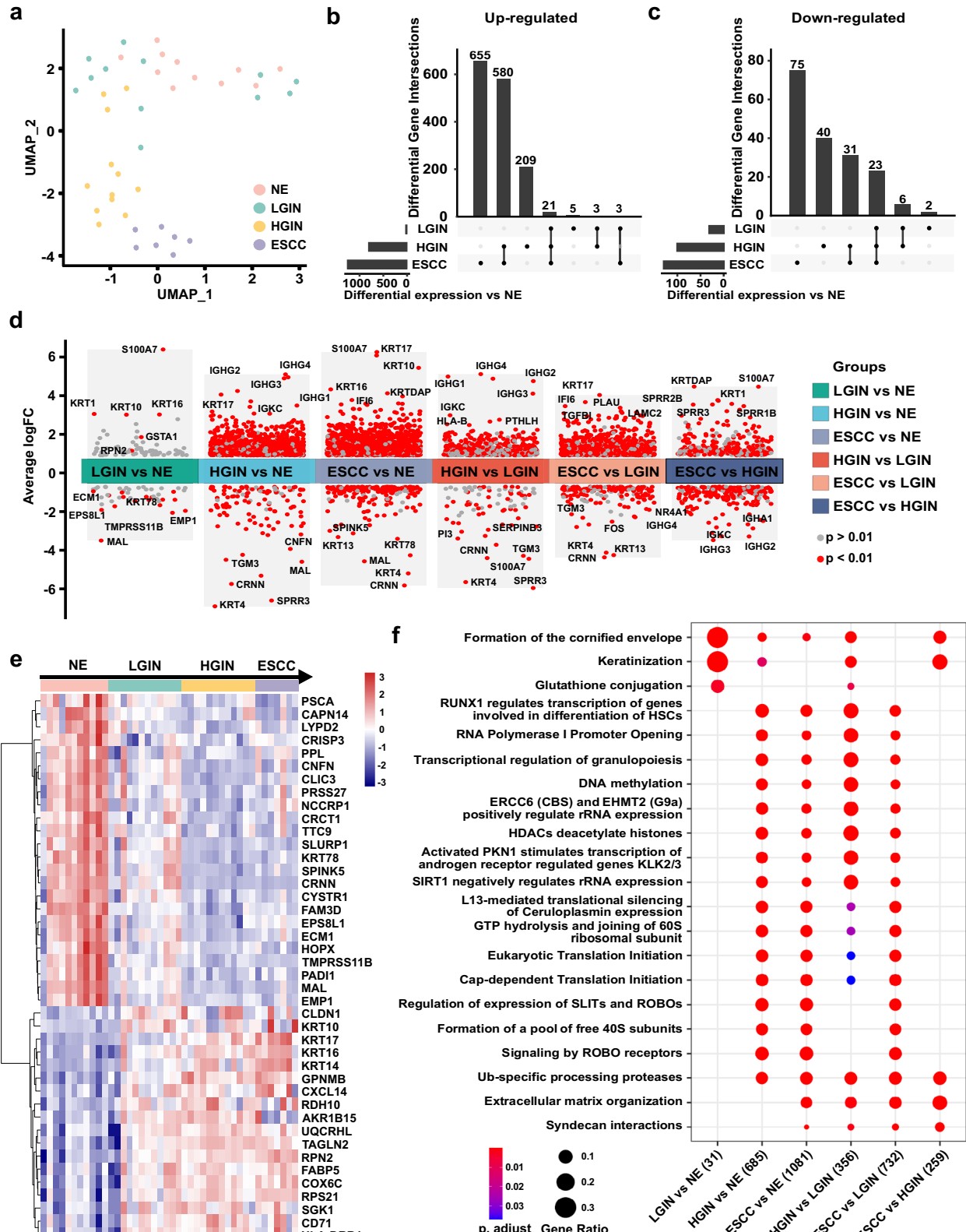

**Fig. 3 | Differentially expressed genes analysis from spatially defined regions of ESPL. a** Plot of principal component analysis (PCA) based on WTA sequencing of spatially defined ESPL (NE, LGIN, HGIN, and ESCC). **b, c** The upset plot of up-regulated and down-regulated DEGs among the NE group, LGIN group, HGIN group, and ESCC group. **d** Comparison of DEGs between different groups (LGIN vs NE; HGIN vs NE; ESCC vs NE; HGIN vs LGIN; ESCC vs LGIN; ESCC vs HGIN). **e** The gene expression heatmap of co-DEGs among LGIN, HGIN, and ESCC groups. **f** The comparison clusters of signaling pathways between different groups. The size of the dots represents the gene expression ratio; the color scale shows the adjusted p value.

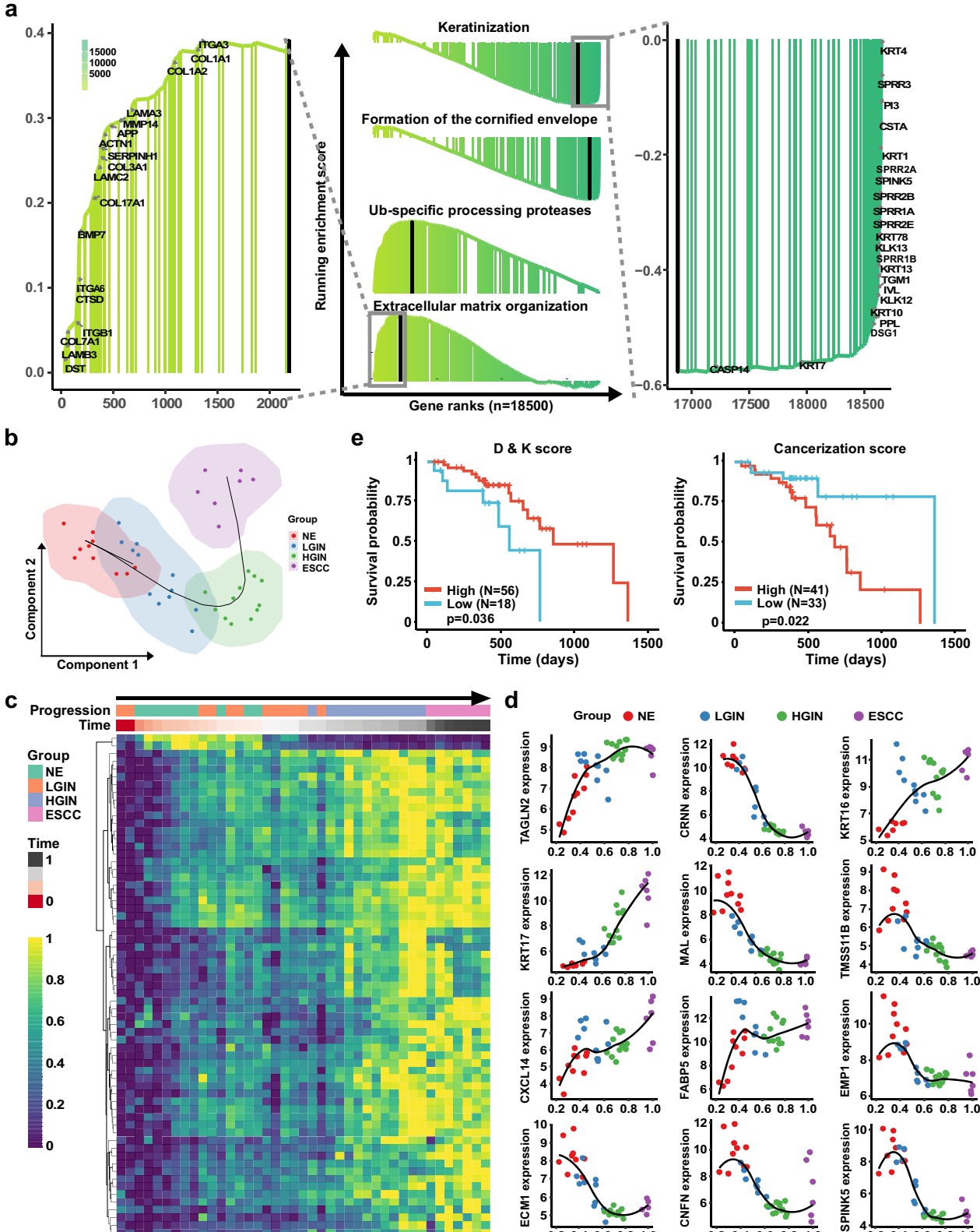

**Fig. 4 | Pseudotime analysis of the esophageal epithelia cell status transitions and prognosis analysis by Differentiation & Keratinization score and Cancerization score. a** Rank-based gene set enrichment analysis of signaling pathway and representative genes. **b** The pseudotime of esophageal epithelia cell transitions showed great correspondence to the ESCC tumorigenesis stages. **c** The heatmap of gradient-developed DEGs according to pseudotime and ESCC progression. **d** Representative gene expression correlated with pseudotime. **e** Overall survival is indicated by Differentiation & Keratinization (D & K) score and Cancerization score in high or low groups of patients.

(nnet), naive bayes (nbs), and decision trees (rpart) are the major algorithms used in machine learning. A prediction model is created to forecast the N, L, and M states by utilizing gene expression data obtained from a particular sample. To assess our prediction model's efficacy in distinguishing N, L, and M states based on co-DEGs (Fig. 5a), we utilized these seven machine learning models. Logloss, also known as logarithmic loss, is a scalar that assesses the classifier's accuracy in predicting class probabilities for classification tasks. It calculates the deviation between predicted and actual target probabilities. Improved algorithm performance is demonstrated by lower logloss values. Multiclass area under the curve (multiclass AUC) is a metric utilized to evaluate classification model performance when handling multiple classes. It measures how well the model can distinguish between these classes. Higher multiclass AUC values indicate that the algorithm is better at distinguishing between different classes. In our study, we sought to ascertain the algorithms that exhibited optimal efficacy in classification tasks by assessing both logloss values and multiclass AUC values. After a thorough evaluation, we identified two algorithms knn and logre that displayed superior performance than other algorithms across both metrics, characterized by a smaller average logloss value and a higher AUC value. Thus, we selected these two algorithms for further investigation of the candidate indicator panel to narrow down the list of candidate genes.

The results of the logre algorithm are presented as a list of genes along with their corresponding importance rank values. The gene list is organized in descending order based on their importance value, from the most significant to the least significant (Supplementary Fig. 4a). To determine the optimal configuration for the knn algorithm, we assessed its logloss value by varying k values and the number of genes utilized. After thorough evaluation, we identified that the optimized result was achieved with a k value of 6 and 20 genes (Supplementary Fig. 4b, c). The gene panel obtained from the knn algorithm is listed in Supplementary Table 2. The overlap between the gene panel generated by the knn algorithm and the genes with an importance rank > 50 from the logre algorithm revealed that *CRNN, KRT17, MAL, KRT16*, and *TAGLN2* genes exhibited higher importance than other genes (Fig. 5b).

Haye et al. reported that KRT17 was detected in HGIN pre-malignant lesions of esophageal mucosa[30]. Additionally, another study indicated that MAL as a regulator of esophageal epithelium differentiation, was observed in normal tissue, but not in dysplastic lesions or cancer tissue[31]. Based on the results obtained from pseudotime analysis, the gene panel of D & K and cancerization score, and the prediction model classifier, we identified *TAGLN2, KRT16*, and *CRNN* as potential candidate indicators. Moreover, we selected *KRT17* and *MAL* as positive controls to assess the effectiveness of our candidate genes in comparison to these previously reported indicators for ESPL diagnosis. Our spatial WTA sequencing data revealed that mRNA levels of *TAGLN2, KRT16*, and *KRT17* increased during the progression of ESCC, while the mRNA levels of *CRNN* and *MAL* genes were down-regulated (Fig. 5c, Supplementary Fig. 5a). Further validation from the TCGA and GTEx database (https://xenabrowser.net) showed that the expression data of these genes in esophageal tumor tissue compared with normal tissue were consistent with our WTA results of the NE and ESCC groups (Supplementary Fig. 5b). These data provide additional evidence to support the credibility and accuracy of our sequencing results. To validate the gene expression, we collected and extracted RNA from paired human normal and ESCC tissue samples, and quantitative RT-PCR was performed. The data demonstrated increased mRNA expression levels of *TAGLN2, KRT16*, and *KRT17*, while *CRNN* and *MAL* exhibited decreased expression. (Fig. 5d, Supplementary Fig. 5c). These findings were consistent with our WTA results and the data obtained from the TCGA database.

To evaluate the feasibility of the candidate genes as indicators of ESPL, we conducted IHC staining on NE, LGIN, HGIN, and ESCC

tissues in the validation cohort (Fig. 5e, f, Supplementary Fig. 5d–f). Our data revealed that the protein level of TAGLN2 significantly increased along with the pathological progression from LGIN to ESCC compared with NE. Similarly, KRT16 expression level was significantly up-regulated in the HGIN stage compared with NE, while CRNN expression was down-regulated in ESPL and ESCC. On the other hand, the protein levels of KRT17 and MAL did not exhibit any significant statistical differences across the four stages. Based on these results, we narrowed down the candidates' list to *TAGLN2, CRNN*, and *KRT16*. To validate our prediction model, we used our IHC data. Our indicator candidates (*TAGLN2, KRT16*, and *CRNN*) demonstrated a high area under the curve (AUC) in distinguishing N, L, and M groups with AUC values of 0.88, 0.853, and 0.952, respectively (Fig. 5g). The prediction model was further validated using data from the GEO dataset (GSE161533) (Fig. 5g). Since the RNAseq data of ESPL is unavailable, we utilized the GEO_GSE161533 data, which includes normal and ESCC samples, for partial validation [https://www.ncbi.nlm.nih.gov/geo/query/acc.cgi?acc=GSE161533]. Our candidate indicators exhibited an AUC over 0.85 in both N and M groups. Among the discovery set, validation set, and GEO set, 131 out of 162 samples were correctly identified. The total accuracy of N, L, and M groups was 47/51, 16/31, and 68/80, respectively (Fig. 5h). The lower predictive accuracy in the validation set of the L group may be attributed to the small number of L group samples and differences in data types (mRNA data of discovery set and GEO set, protein expression data of validation set). Nevertheless, these data suggest that *TAGLN2, KRT16*, and *CRNN* may have potential utility as diagnostic markers for ESPL, even during the LGIN stage.

## Verification of the association between candidate genes and ESCC progression

To investigate the association of these candidate indicators with progression, we obtained paired tissue samples from patients who were initially diagnosed with LGIN and later progressed to HGIN, or those who were first diagnosed with HGIN and later progressed to ESCC for IHC validation. However, due to the rarity of paired tissue samples from the same individual corresponding to before- and after-progression, as well as the typically several years required for LGIN progression to ESCC, paired tissue samples from individuals who progressed from LGIN to ESCC were unavailable. We labeled the tissue taken from the initial diagnosis as "before-progression" (Before) and the tissue procured from later diagnoses as "after-progression" (After). Our analysis of the IHC scores on all paired progressed samples revealed that TAGLN2 protein levels were significantly increased in the after-progression group tissues compared with the before-progression group, while CRNN expression decreased (Fig. 6a). Further analysis of the IHC scores in LGIN progression and HGIN progression samples showed that TAGLN2 IHC score was significantly up-regulated in the after-progression group of LGIN samples. In the HGIN progression samples, the mean IHC score value of TAGLN2 exhibited an increase in the after-progression group compared to the before-progression group, although it did not reach statistical significance due to the limited sample size (Fig. 6b–d). On the other hand, CRNN's IHC score significantly decreased in both LGIN and HGIN progression samples (Fig. 6b–d). These results strongly suggest that TAGLN2 and CRNN are correlated with ESCC progression. However, KRT16 did not show any statistical significance in these paired samples, indicating that it cannot be used as an indicator of ESCC progression (Supplementary Fig. 6a–c). Therefore, KRT16 was excluded from further studies.

## Candidate genes expression in epithelial cell type, pan-cancer, and prognosis analysis

Single-cell RNA sequencing (scRNA-seq) provides a powerful tool for studying tumor heterogeneity and the tumor microenvironment.

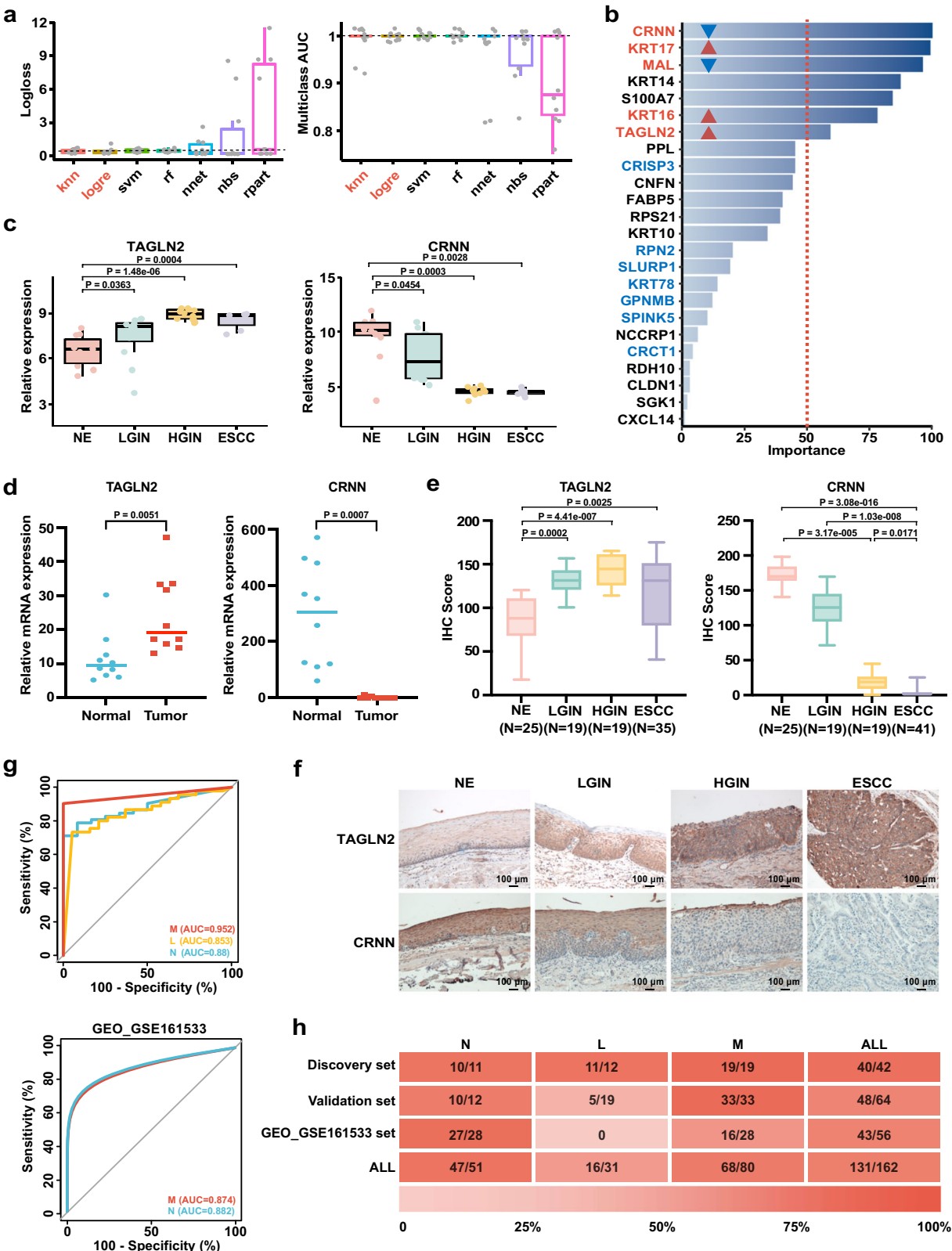

Considering ESPL is mainly localized in the epithelium layer, we wondered whether these candidate genes are specifically expressed in epithelial cells and the expression level. Fresh tissue samples are a prerequisite for performing scRNA-seq analysis. However, as surgical procedures are not standard clinical practice for patients diagnosed with LGIN, biopsy specimens cannot provide enough fresh tissue for scRNA-seq. Thus, we performed bioinformatics analysis using scRNA-

seq data from published studies to evaluate *TAGLN2* and *CRNN* gene expression in normal and ESCC tissue (Fig. 7a). For this analysis, we utilized the scRNA-seq datasets GSE159929 and GSE160269. GSE159929 contains a comprehensive adult human cell atlas with 84,363 cells from 15 different tissue organs of a single adult donor. GSE160269 consists of 208,659 single-cell transcriptomes from 60 individuals diagnosed with ESCC, including 60 ESCC tumors and 4

**Fig. 5 | Potential indicators screening for the diagnosis of ESPL and predicting the risk of ESCC. a** Logloss and multiclass AUC were analyzed by seven machine learning models (n = 10 for NE, LGIN, HGIN, and ESCC group, respectively). **b** The importance rank of signatures. The overlap genes between logre and knn algorithm results are marked with red and blue colors. Red color-marked genes indicated the overlap genes with importance rank >50. The red and blue triangles indicated gene expression up-regulated and down-regulated, respectively. knn (k-nearest neighbor); logre (logistic regression); svm (support vector machine); rf (random forest); nnet (neural network regression); nbs (naive bayes); rpart (decision trees). **c** Candidate gene (*TAGLN2* and *CRNN*) expression in the four pathological stages by spatial WTA analysis. NE, LGIN, HGIN, and ESCC ROIs number (n = 11, 12, 12, 7). Mann–Whitney U test was used for the comparison with the NE group. In the box plots (**a, c**), the boxplot shows the median (central line), upper and lower quartiles (box limits), and 1.5 × interquartile range (whiskers). **d** Relative mRNA expression of

candidate genes in paired normal and ESCC tissue. NE (n = 10), ESCC tumor tissue (n = 10) biologically independent samples. The data was analyzed by a two-tailed paired t-test. **e** Positive staining statistics of TAGLN2 and CRNN in the four pathological stages by IHC staining. The sample size is labeled in the figure. Kruskal–Wallis test and corrected by Dunn's test for multiple comparisons. The boxplot shows the median (central line), upper and lower quartiles (box limits), and min to max range (whiskers). **f** Representative pictures of IHC stained slides (Scale bar, 100 μm, representative of n = 3 independent experiments.). **g** ROC plots of gene panel (*TAGLN2*, *KRT16*, *CRNN*) in distinguishing N and M groups from IHC staining data and GEO dataset (GSE161533). **h** Overall performance metrics of the prediction model for N, L, and M groups. Graduated colors indicate accuracy levels. The number in each box indicate correctly identified samples/total sample number. Source data are provided as a Source Data file.

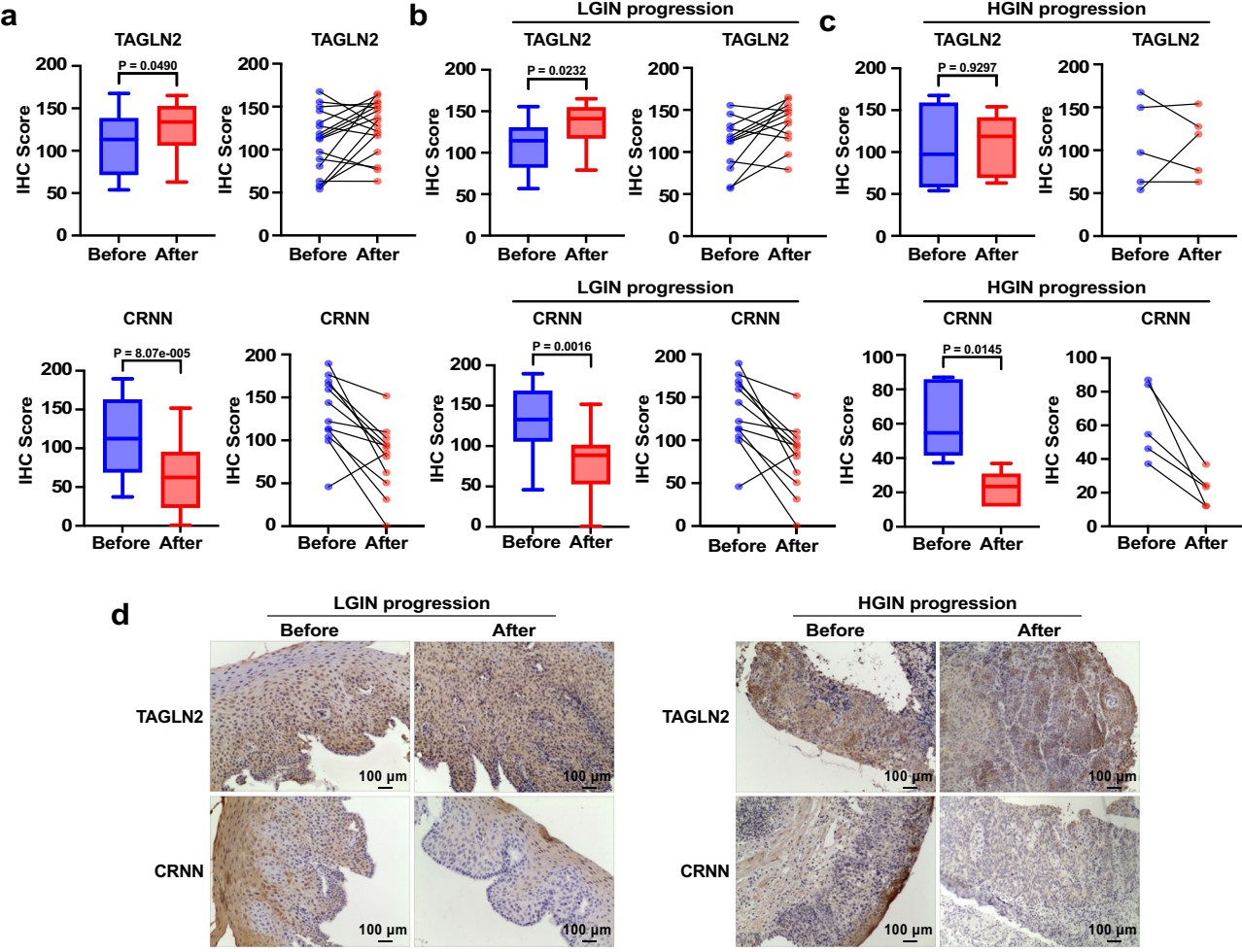

**Fig. 6 | Verification of the association between candidate genes and ESCC progression. a** An analysis of the IHC scores of TAGLN2 and CRNN was conducted on all paired progressed samples (including LGIN progression samples (n = 12) and HGIN progression samples (n = 5) biologically independent samples). **b** Statistics of the IHC scores of TAGLN2 and CRNN in paired LGIN progression (n = 12) tissue. **c** Statistics of the IHC scores of TAGLN2 and CRNN in paired HGIN progression

samples (n = 5) tissue. **d** Representative images of TAGLN2 and CRNN in paired LGIN progression and HGIN progression samples (Scale bar, 100 μm). In the box plots (**a, b, c**), the boxplot shows the median (central line), upper and lower quartiles (box limits), and min to max range (whiskers) analyzed by a two-tailed paired t-test. Source data are provided as a Source Data file.

adjacent normal tissue samples. We specifically focused on the epithelial cells data from both datasets and excluded samples with less than 1000 cells. The remaining epithelial cells were further divided into basal squamous epithelial cells and squamous epithelial cells. Our analysis revealed that *TAGLN2* was expressed in both types of epithelial cells and exhibited higher expression levels in cancer tissue compared to the normal group. On the other hand, *CRNN* showed specific

expression in squamous epithelial cells of normal tissue and negligible expression in cancer tissue. Based on our findings, *TAGLN2* and *CRNN* are expressed in squamous epithelial cells, which suggests their potential utility as candidate indicators for both ESPL and ESCC diagnoses. This observation aligns with our IHC results. However, it is also possible that these candidate genes contribute to the acceleration of ESCC progression by triggering the transformation of normal

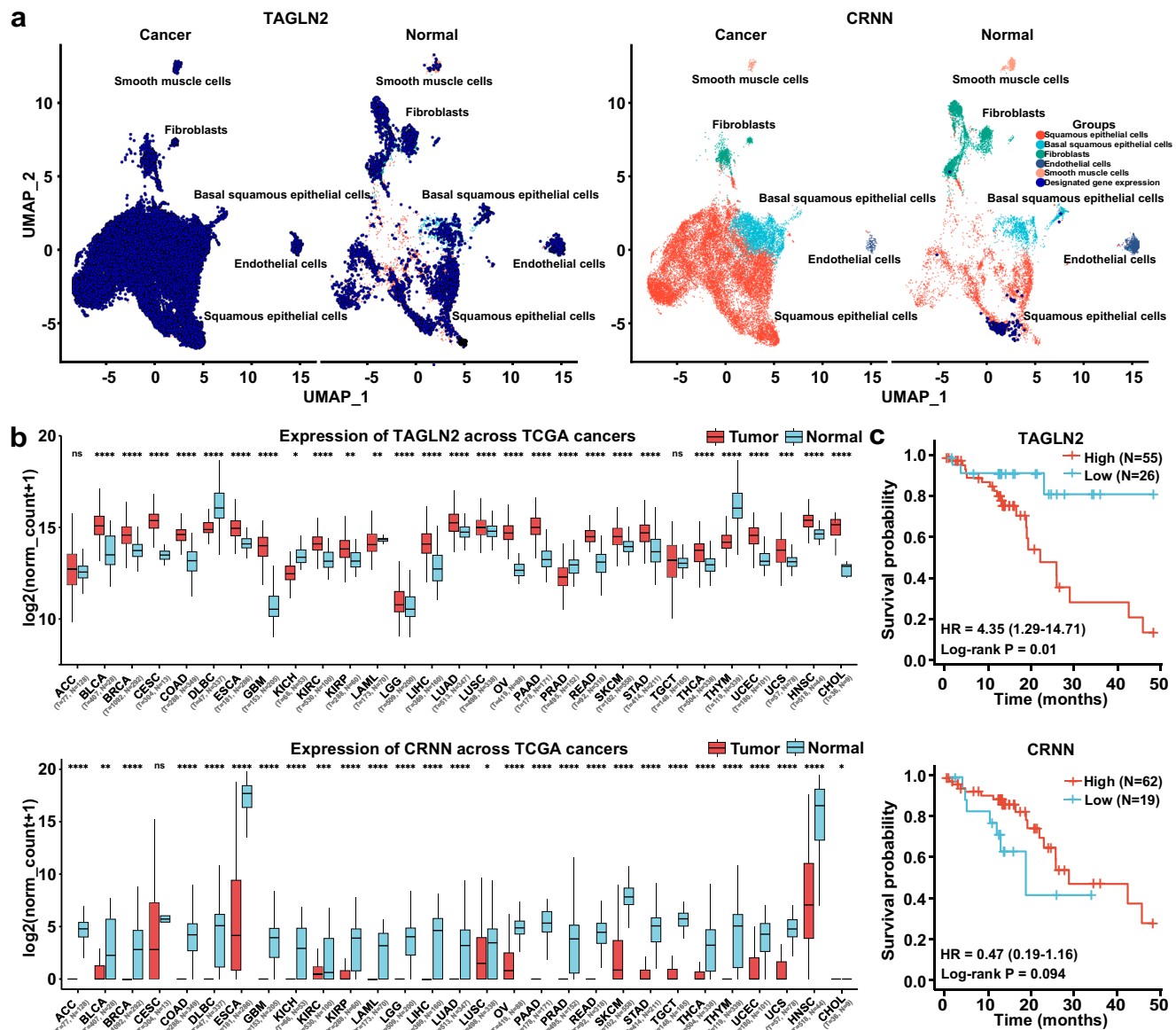

**Fig. 7 | Candidate genes expression in epithelial cell type, pan-cancer, and prognosis analysis. a** Bioinformatics analysis of *TAGLN2* and *CRNN* gene expression in normal and ESCC tissue by single-cell sequencing. Different colors indicated different cell types. Dark blue dots represented the designated gene expression **b** Candidate gene expression in multi-cancer types from TCGA database. The sample size is indicated in the figure. The boxplot shows the median (central line), upper and lower quartiles (box limits), and 1.5 × interquartile range (whiskers). For the comparison of *TAGLN2* expression between Tumor and Normal group: ACC (p = 0.3045), BLCA (p = 2.53E−08), BRCA (p = 8.16E−92), CESC (p = 8.90E−09), COAD (p = 2.52E−53), DLBC (p = 3.09E−27), ESCA (p = 1.38E−24), GBM (p = 7.06E −97), KICH (p = 0.0144), KIRC (p = 1.97E−10), KIRP (p = 0.0021), LAML (p = 0.0037), LGG (p = 3.91E−07), LIHC (p = 5.23E−36), LUAD (p = 1.27E−18), LUSC (p = 4.70E−06), OV (p = 1.07E−91), PAAD (p = 1.21E−33), PRAD (p = 3.76E−23), READ (p = 1.51E−46),

SKCM (p = 1.94E−10), STAD (p = 1.68E−18), TGCT (p = 0.5421), THCA (p = 6.53E−26), THYM (p = 2.71E−76), UCEC (p = 1.42E−49), UCS (p = 0.0002), HNSC (p = 8.57E−13), CHOL (p = 4.64E−13). For the comparison of *CRNN* expression between Tumor and Normal group: ACC (p = 1.58E−72), BLCA (p = 0.0081), BRCA (p = 2.60E−42), CESC (p = 0.155), COAD (p = 3.13E−111), DLBC (p = 2.79E−63), ESCA (p = 7.67E−70), GBM (p = 1.41E−55), KICH (p = 3.45E−10), KIRC (p = 0.0005), KIRP (p = 1.08E−12), LAML (p = 2.61E−16), LGG (p = 4.71E−59), LIHC (p = 2.74E−35), LUAD (p = 5.88E−59), LUSC (p = 0.0355), OV (p = 2.59E−30), PAAD (p = 5.26E−76), PRAD (p = 3.54E−32), READ (p = 5.80E−131), SKCM (p = 4.17E−36), STAD (p = 9.16E−35), TGCT (p = 6.33E−88), THCA (p = 4.27E−63), THYM (p = 2.68E−83), UCEC (p = 7.22E−17), UCS (p = 2.10E −40), HNSC (p = 8.01E−12), CHOL (p = 0.0317) (Mann–Whitney U test). **c** Overall survival analysis of candidate genes from TCGA database. Source data are provided as a Source Data file.

epithelial cells into malignant ones. Further research and functional studies are warranted to fully understand the roles of *TAGLN2* and *CRNN* in esophageal cancer development and progression.

To assess the specificity of candidate gene expression, we investigated their expression across multiple cancer types using data from the TCGA and GTEx databases (https://xenabrowser.net) (Fig. 7b). The results revealed that *TAGLN2* was broadly expressed in various cancer types, with significantly increased expression especially in tumor tissues of GBM (Glioblastoma Multiforme), CHOL

(Cholangiocarcinoma), and ESCA (Esophageal Carcinoma) compared to normal tissues. On the other hand, *CRNN* expression significantly decreased in tumor tissues of DLBC (Diffuse Large B-Cell Lymphoma), HNSC (Head and Neck Squamous Cell Carcinoma), and ESCA. To assess the relationship between gene expression and cancer prognosis, we conducted overall survival analysis for the candidate genes (http://www.kmplot.com). Higher mRNA expression of *TAGLN2* or lower mRNA level of *CRNN* was associated with shorter overall survival and poorer prognosis (Fig. 7c).

## Identifying candidate indicators' role in accelerating ESCC progression

Functional investigations can offer initial insights into the molecular mechanisms underlying the influence of genes in promoting cancer progression. The functions of *TAGLN2* and *CRNN* in different types of cancer have been studied in various literature[32–34]. With the intent to estimate the impact of candidate genes in promoting ESCC progression, we constructed *TAGLN2* knockdown cell lines and *CRNN* overexpressed cell lines to evaluate the effects of these candidate genes in ESCC. First, we detected the protein level of TAGLN2 and CRNN in ESCC cell lines (Supplementary Fig. 7a) to select high-expression cell lines for *TAGLN2* knockdown analysis and gene low-expression cell lines for *CRNN* overexpression study. The efficiency of *TAGLN2* knockdown and *CRNN* overexpression was assessed by Western blot (Fig. 8a). *TAGLN2* knockdown or *CRNN* overexpression inhibited ESCC cells proliferation (Fig. 8b). The suppression of colony formation in ESCC was observed due to *TAGLN2* knockdown or *CRNN* overexpression through the anchorage-independent cell growth assay and clonogenic formation assay (Fig. 8c–f).

Tumor organoids are valuable tools in cancer research, facilitating gene function studies and anti-cancer drug screening[35]. *TAGLN2* knockdown or *CRNN* overexpression led to a reduction in organoid numbers compared to the relevant vector control group (Fig. 8g, h).

Patient-derived xenograft (PDX) model potentially retains the donor's genetic characteristics, making it more similar to human than cell line-derived xenograft. To investigate the role of *TAGLN2* and *CRNN* in ESCC progression, we selected a PDX case (LEG404) with higher TAGLN2 expression and lower CRNN expression from our established ESCC PDX models (Supplementary Fig. 7b). Subsequently, we transplanted the PDX tumor into mice and administered designated candidate gene viruses twice a week to evaluate the impact of these genes on ESCC growth in vivo. Our data revealed that *TAGLN2* knockdown or *CRNN* overexpression inhibited ESCC PDX tumor growth, leading to reduced tumor volume and tumor weight, without any adverse effects on the mice body weight (Fig. 8i–k, Supplementary Fig. 7c).

To ensure the reliability of our findings, we performed a rescue experiment to address potential off-target effects and unphysiologically high amplification levels resulting from shRNA and gene overexpression techniques. The rescue experiment involved overexpressing *TAGLN2* in cells where *TAGLN2* was knocked down, or inhibiting *CRNN* where *CRNN* was overexpressed. Western blot analysis confirmed the efficiency of the rescue experiment (Supplementary Fig. 8a). In the rescue group, we observed increased cell proliferation and colony formation, as indicated by cell proliferation assay, anchorage-independent cell growth assay, and clonogenic formation assay, upon manipulating *TAGLN2* and *CRNN* (Supplementary Fig. 8b–f). Collectively, the in vitro and in vivo data support the notion that *TAGLN2* and *CRNN* participated in accelerating malignancy in ESCC cells.

Furthermore, we conducted experiments using a human normal esophageal cell line (SHEE) to investigate the role of these genes in the transformation from normal cells to malignant cells. Specifically, we examined the effects of overexpressing *TAGLN2*, which is lowly expressed in SHEE, and knocking down *CRNN*, which is highly expressed in SHEE. The successful overexpression of *TAGLN2* and knockdown of *CRNN* were confirmed by Western blot analysis (Fig. 9a). Under normal conditions, SHEE cells are unable to form colonies in anchorage-independent cell growth assays. However, when these cells were stimulated with EGF, they exhibited signs of transformation into malignant cells, as evidenced by the formation of clones (Fig. 9b). By simulating the process of normal cells transforming into malignant cells with the addition of EGF, we aimed to evaluate whether *TAGLN2* and *CRNN* accelerate the progression of ESCC. The results from cell proliferation experiments revealed that overexpression of *TAGLN2* or

knockdown of *CRNN* significantly promoted SHEE cell proliferation (Fig. 9c). In anchorage-independent cell growth assays, both *TAGLN2* overexpression and *CRNN* knockdown significantly increased the number of colonies compared to the control group, indicating that *TAGLN2* promotes the transformation of normal cells to malignant cells, while *CRNN* inhibits this transformation (Fig. 9d, e). Collectively, these findings, along with the results of IHC analysis of paired samples before- and after-progression and the normal esophageal cell transformation assay, suggest that *TAGLN2* and *CRNN* are associated with the progression of ESCC. Furthermore, *TAGLN2* is implicated in promoting ESCC progression, while *CRNN* is involved in inhibiting ESCC progression by regulating cell proliferation.

Although most patients with LGIN do not require immediate treatment, monitoring the expression levels of TAGLN2 and CRNN can serve as indicators for potential progression to ESCC at ESPL stages. Patients displaying high-expression levels of TAGLN2 and low-expression levels of CRNN should be closely monitored for ESCC development, as this information can contribute to early diagnosis and prevention strategies for ESCC.

## Discussion

Advances in tissue omics analysis, such as genomics, epigenomics, and next-generation sequencing techniques, have provided significant opportunities for biomarker discovery. However, current ESCC-related biomarkers based on gene expression differences between normal and ESCC tissues have limited prognostic predictive ability. To address this limitation and identify indicators for the detection of ESPL and prediction of progression risk, we employed spatial whole-transcriptome analysis (WTA) using FFPE samples. This approach allows us to sequence limited biopsy tissue samples and specifically target the ESPL region compared to bulk RNAseq. Nano-string WTA technology offers an unbiased landscape of expressed human whole transcripts at designated tissue sections, surpassing the limitations of in situ hybridization or immunohistochemistry with limited antibodies or probes. Nonetheless, spatial WTA has some inherent limitations, such as the maximum size of the region of interest (ROI) being 800 μm, which may not cover the entire tissue area, and each ROI containing approximately 600 cells, preventing single-cell resolution. Furthermore, unlike common 10X Visium spatial transcriptome sequencing, nanostring WTA cannot reconstruct a gene expression heatmap at each spot of the tissue. Nevertheless, the probe-based detection method of spatial WTA analysis provides high sensitivity to detect lowly expressed genes. Moreover, several tissues can be placed on one slide during the same operation, reducing the impact of human manipulation on gene expression. Despite these limitations, spatial WTA remains a valuable approach for researchers to study spatial-related gene expression in the context of ESPL and ESCC.

Single-cell RNA sequencing (scRNA-seq) has emerged as a powerful tool in recent years for studying tumor heterogeneity and tumor microenvironments. By analyzing specific gene markers' expression levels, researchers can identify and categorize different cell subtypes, allowing for the study of gene expression, related signaling pathways, and even cell fate within these subtypes. Additionally, scRNA-seq has the potential to uncover previously unidentified cell subtypes. However, conducting scRNA-seq requires fresh tissue, which is not typically obtained through surgery for LGIN patients in clinical practice. As a result, biopsy specimens may not be sufficient for scRNA-seq analysis. Currently, human scRNA-seq data related to ESCC mainly consists of normal/adjacent tissue and ESCC tissue samples[36,37]. Furthermore, scRNA-seq analysis of continuous tumorigenic lesions has primarily been performed on mouse models[38], which introduces species variation between humans and mice. To characterize our candidate indicators more reliably, we utilized human scRNA-seq data to investigate *TAGLN2* and *CRNN* distributions in normal and cancerous epithelial cells. Our analysis revealed distinct expression patterns for *TAGLN2*

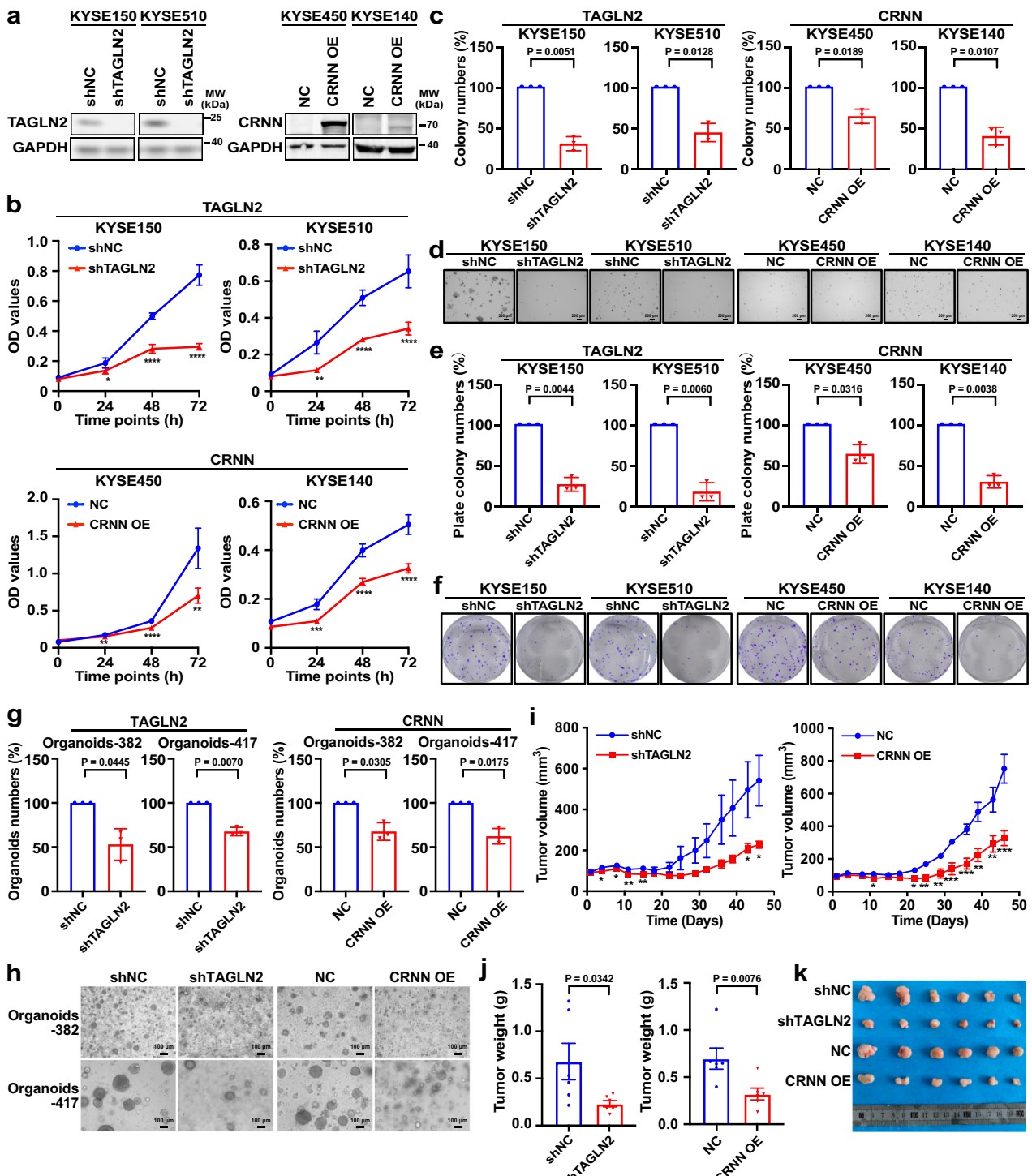

**Fig. 8 | Identifying candidate genes' role in accelerating ESCC progression.**
**a** Knockdown and overexpression efficiency of *TAGLN2* and *CRNN* detected by western blot. **b** Cell proliferation analysis of *TAGLN2* silenced or *CRNN* over-expressed cells by MTT assay (mean ± SD, n = 6 biological replicates). *p = 0.0109, ****p = 3.73e−08, ****p = 3.23e−006 for KYSE150; **p = 0.0015, ****p = 3.09e−005, ****p = 1.22e−005 for KYSE510; ** p = 0.0018, ****p = 3.13e−008, **p = 0.0038 for KYSE450; ***p = 0.0004, ****p = 9.01e−007, ****p = 1.76e−006 for KYSE140 by a two-tailed unpaired t-test at 24 h, 48 h, and 72 h, respectively. **c** *TAGLN2* silenced or *CRNN* overexpressed cells colony formation ability assessed by anchorage-independent cell growth assay. **d** Representative images of anchorage-independent cell growth assay (Scale bar, 200 μm). **e** Colony formation ability was analyzed by clonogenic formation assay. **f** Representative images of clonogenic formation assay. **g** Inhibitory effects of organoid growth by knocking down *TAGLN2* or overexpressing *CRNN*. **h** Representative pictures of organoids after knocking down

*TAGLN2* or overexpressing *CRNN* (Scale bar, 100 μm). **i, j** Tumor volume and tumor weight are decreased by knocking down *TAGLN2* or overexpressing *CRNN* (n = 6 mice). **k** Pictures of tumors after mice euthanasia at 46 days. Tumor volume and tumor weight are shown as mean values ± SEM by a one-tailed unpaired t-test (p value for shTAGLN2 tumor weight with Welch's correction). For Tumor volume comparison of shTAGLN2 and shNC: day 4 (*p = 0.0392), day 8 (*p = 0.0379), day 11 (**p = 0.0030), day 15 (**p = 0.0037), day 43 (*p = 0.0330), day 46 (*p = 0.0168). For Tumor volume comparison of CRNN and NC: day 11 (*p = 0.0171), day 22 (*p = 0.0128), day 25 (**p = 0.0021), day 29 (**p = 0.0028), day 32 (***p = 0.0008), day 36 (***p = 0.0005), day 39 (**p = 0.0022), day 43 (**p = 0.0070), day 46 (***p = 0.0009). The data (**c, e, g**) are shown as percentages compared with the control group by a two-tailed unpaired t-test with Welch's correction (n = 3, independent experiments, mean ± SD). Source data are provided as a Source Data file.

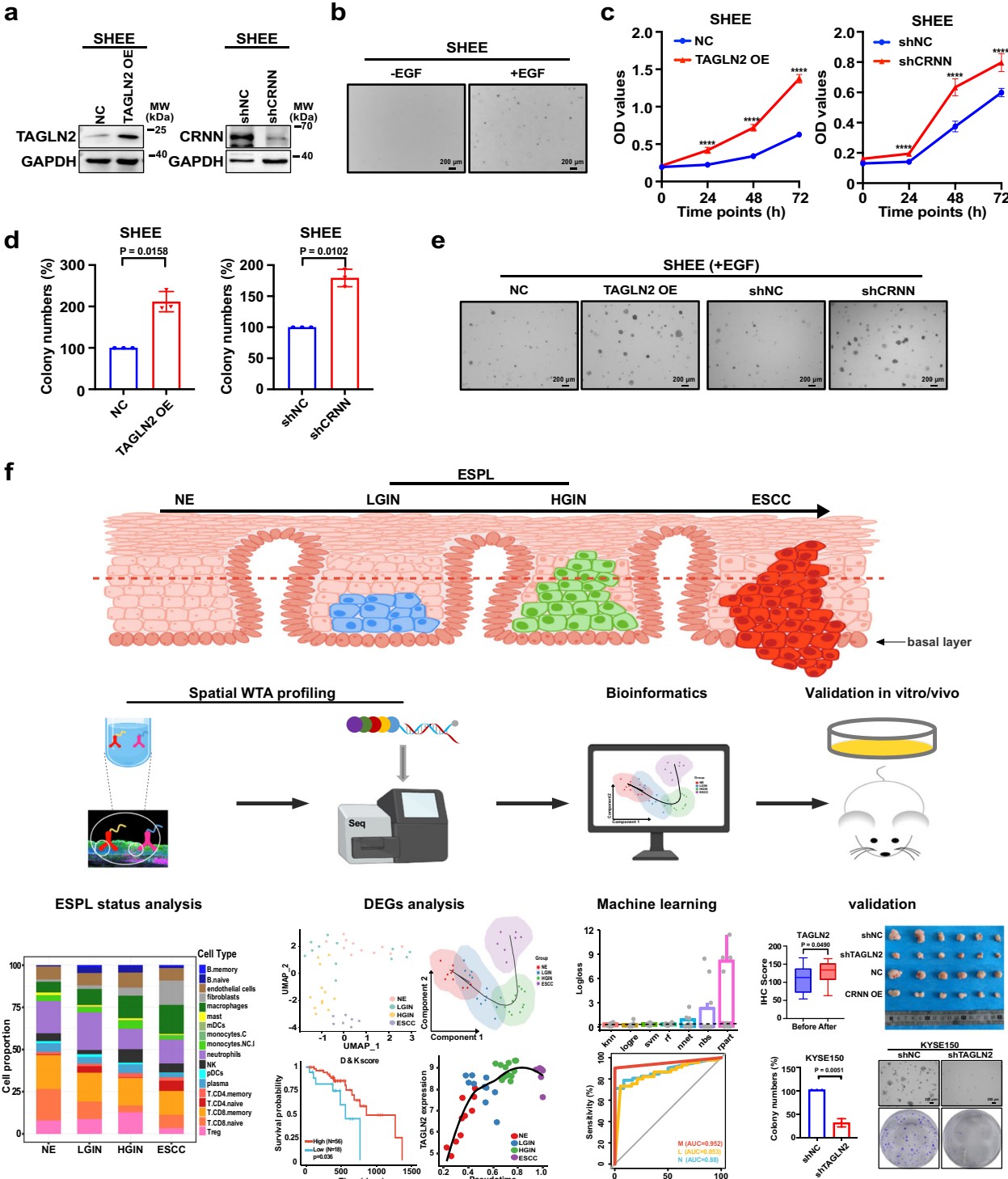

**Fig. 9 | Identifying candidate genes' role in the transformation of human normal esophageal cells to malignant cells and the schematic of the spatial WTA analyses of ESPL and ESCC. a** Overexpression and knockdown efficiency of *TAGLN2* and *CRNN* detected by western blot. **b** Colony formation ability of human normal esophageal cell (SHEE) with the stimulation of EGF by anchorage-independent cell growth assay (Scale bar, 200 μm). **c** Cell proliferation analysis of *TAGLN2* overexpression or *CRNN* knockdown in normal esophageal cells (n = 6 biological replicates, mean ± SD) . ****p = 1.32e-006, ****p = 4.30e-009, ****p = 1.21e−010 for the comparison of TAGLN2 OE and NC; ****p = 5.59e−007, ****p = 2.50e-006, ****p = 2.23e−005 for the comparison of shCRNN and shNC by a two-tailed unpaired t-test at 24 h, 48 h, and 72 h, respectively. **d** Colony formation ability of *TAGLN2* overexpressed or *CRNN* silenced SHEE cells with the stimulation of EGF analyzed by anchorage-independent cell growth assay (n = 3, independent experiments by a two-tailed unpaired t-test with Welch's correction, mean ± SD). **e** Representative image of anchorage-independent cell growth assays (Scale bar, 200 μm). **f** Schematic of the spatial WTA analyses of ESPL and ESCC. Human NE, LGIN, HGIN, and ESCC tissue were analyzed by spatial WTA profiling. The status of ESPL, DEGs, and the prediction model were analyzed by bioinformatics. Candidate genes were validated for the association with ESCC progression in paired before-progression and after-progression tissue samples from the same individuals. ESCC cell lines, normal esophageal cells, organoids, and the ESCC PDX model were used for identifying the mechanism of candidate genes' role in accelerating ESCC progression. Data are presented as mean values ± SD by a two-tailed paired t-test. The boxplot shows the median (central line), upper and lower quartiles (box limits), and min to max range (whiskers). Scale bar, 200 μm. Source data are provided as a Source Data file.

and *CRNN* in these cell types, with *CRNN* showing specific distribution in normal squamous epithelial cells.

Whole-genome and whole-exome sequencing provide comprehensive insights into genome variations in esophageal cancer. Common alterations reported include *TP53*, *NOTCH1*, and *PIK3CA*[39–41]. Dysplasia, the precursor lesion of ESCC has been found to harbor similar driver genes as ESCC itself[8]. Copy number alterations have been observed to persist from dysplasia to ESCC. However, these alterations are not specific to esophageal cancer, as evidenced by the high mutation rate of *TP53* in ovary cancer and colon cancer[42]. Unlike other cancers with specific mutations such as *BRCA1/2* in breast cancer and *EGFR* in lung cancer, ESCC lacks tumor-specific mutations. Notably, somatic mutations also occur in physiologically normal human esophageal tissue[43]. Thus, we consider mutations may not be indispensable for ESCC initiation and progression. Transcriptome aberrance offers an alternative explanation for these phenomena. Our data reveals that the number of DEGs in the LGIN group (63) is considerably lower than that in the HGIN group (826) and ESCC group (1298). This observation suggests a potential relationship between DEGs in LGIN samples and ESCC initiation and development, particularly during the first step from NE to LGIN transition. Consequently, we investigated the expression of these candidate initiators in the four pathological stages and evaluated their function in ESCC progression.

The publication using continuous tumorigenic lesions of mice for scRNA-seq identified six transitional-related genes selected from different epithelial clusters[38]. Although our WTA data showed similar expression patterns for these six genes, some differences were observed (Supplementary Fig. 9a). Notably, *S100A8* and *ALDH3A1* did not demonstrate statistical significance in our data, despite the publication reporting a notable decrease in cancer incidence. *TOP2A*, *ATF3*, and *MMP14* showed no statistical significance in the LGIN group compared with NE, indicating their unsuitability for LGIN diagnosis. Additionally, *ITGA6*, which was significantly up-regulated during ESCC progression, was excluded from our candidate list based on stricter criteria ($|\log2(FC)| \geq 1$ and $p < 0.05$). We also analyzed other indicators previously utilized for ESPL diagnosis in clinical and preclinical settings[44–51] (Supplementary Fig. 9b). In the LGIN stage, none of these markers significantly increased compared with NE. However, *EGFR* and *LAMC2* exhibited marked elevation in HGIN and ESCC, suggesting their potential in identifying HGIN and ESCC. *TP53*, *MKI67*, and *TLR9* showed differences in only one stage, while *ANO1*, *USP9X*, *CCN2*, and *LTO1* did not show any differences across stages. Collectively, our candidate indicators demonstrated superior performance compared to the reported indicators.

Yao and colleagues conducted a comprehensive scRNA sequencing study on mice with multistep tumorigenic lesions[38], revealing the FibC8 fibroblast cluster with increased expression of genes from Myc and angiogenesis pathways as ESCC tumorigenesis advanced, with the highest proportion observed at the ESCC stage. Similarly, in our study, we also observed an enrichment of angiogenesis-related genes in the ESCC stage. In their study, a reduction in the proportion of CD8+ memory T cells was observed as the tumorigenic process progressed beyond the INF stage, indicating a dominant non-effective CD8+ T cell microenvironment during the precancerous stages. Our study also demonstrated a similar decrease in CD8+ memory T cells at ESPL stages (Fig. 2c). The consistency of these findings between the mice model and humans suggests that the mouse model induced by 4NQO is a suitable approach for simulating the progression of esophageal cancer.

In summary, our study utilized a spatial whole-transcriptome atlas to identify potential indicators for predicting ESCC risk at ESPL stages and to investigate the underlying mechanism of pathogenesis from ESPL to ESCC transition (Fig. 9f). We analyzed the biological processes, regulatory network, abnormal metabolic pathways, signal transduction pathways, and immune-related pathways that occur during tumorigenesis in ESPL and ESCC stages. Our findings indicated an immunosuppressive condition in ESPL stages, potentially contributing to the acceleration of ESCC tumorigenesis. Through spatial WTA analysis and IHC staining, we observed an upregulation of TAGLN2 and a downregulation of CRNN expression during ESCC progression from ESPL to ESCC. Notably, TAGLN2 expression significantly increased in the paired after-progression tissues compared with before-progression tissue from the same individual, while CRNN expression decreased, suggesting a correlation of TAGLN2 and CRNN with ESCC progression. Functional studies involving *TAGLN2* knockdown and *CRNN* overexpression demonstrated their inhibitory effects on ESCC cell proliferation, colony formation, organoid growth, and ESCC PDX tumor growth. Furthermore, we observed that *TAGLN2* overexpression and *CRNN* knockdown promoted the transformation of normal esophageal cells into malignant cells. This provides compelling evidence that *TAGLN2* is indeed involved in promoting ESCC progression, while *CRNN* exerts an inhibitory role in ESCC progression through the regulation of cell proliferation. The insights gained from our study provide fundamental information for understanding the pathological process of ESCC development and may serve as an early warning for ESCC, contributing to the prevention and early intervention of esophageal cancer.

## Methods
### Experimental reagents
KRT16 (66802-1-Ig, Clone No. 2H4D8, 1:2000 for WB, 1:1000 for IHC), KRT17 (17516-1-AP, 1:1000 for WB, 1:50 for IHC), TAGLN2 (10234-2-AP, 1:1000 for WB, 1:100 for IHC), CRNN (11799-1-AP, 1:1000 for WB, 1:200 for IHC) antibodies were purchased from Proteintech company. MAL (MA5-32924, Clone No. B5-G3, 1:1000 for WB, 1:100 for IHC) was obtained from the Invitrogen company. CD68 (ab955, Clone No. KP1, 1:50 for IF) and alpha-SMA (ab124964, Clone No. EPR5368, 1:200 for IF) were purchased from Abcam company. GAPDH (TA-08, Clone No. OTI2D9, 1:2000 for WB) was obtained from the ZSGB-BIO company. Goat anti-rabbit IgG H & L (HRP) (ab205718, 1:5000 for WB) antibody was purchased by Abcam company.

### Spatial whole-transcriptome atlas
Five micrometers slices of ESPL and ESCC patient formalin-fixed paraffin-embedded (FFPE) tissue samples were sent to Nano-String spatial whole-transcriptome analysis (Fynn Biotechnologies Ltd). The tissue samples utilized for spatial WTA sequencing were obtained from patients diagnosed with LGIN, HGIN, or ESCC (n = 6, 6, and 7, respectively). We opted not to select surgical samples from ESCC patients that contained LGIN and HGIN adjacent to the cancerous area. This decision was driven by the fact that the patients had already been diagnosed with ESCC. Consequently, it is unclear whether there are alterations in the gene or transcriptome information of the LGIN and HGIN adjacent to the cancerous area when compared to patients with LGIN and HGIN under normal conditions without a history of ESCC. Specifically, the NE group samples were selected from the normal epithelial region of LGIN patients, indicating that NE and LGIN samples originate from the same person (n = 6). To ensure experimental accuracy, a total of four slides were employed for spatial WTA sequencing. Slides 1–3 contained ESPL tissues, each slide consisting of two LGIN and two HGIN patient tissues. Slide-4 comprised an ESCC tissue microarray (TMA), with each spot corresponding to one ESCC patient sample. Each ESPL slide included different sample types (NE, LGIN, and HGIN), which effectively minimized potential slide effects. Given the larger area of ESPL samples, each LGIN sample was subjected to the selection of 3 to 4 ROIs, including 2 ROIs from the LGIN region and 1 to 2 ROIs from the NE region. Each HGIN sample was subjected to the selection of 2 ROIs from the HGIN region. In contrast, due to the smaller tissue area of ESCC tissue samples, only one ROI was selected per patient sample. Ultimately, a total of 11 ROIs for NE, 12 ROIs for

LGIN, 12 ROIs for HGIN, and 7 ROIs for ESCC were collected and sequenced using spatial whole-transcriptome profiling. All sequenced ROIs' data were incorporated into further analysis. All the samples were evaluated and confirmed by a pathologist from the Affiliated Cancer Hospital of Zhengzhou University (Zhengzhou, China). The samples of this study were approved by the Ethics Committee of the same institute. Written informed consent was provided by each patient before any investigation was conducted.

The FFPE samples underwent dewaxing and hydration following the DSP standard SOP. Immunohistochemical antigen repair was conducted using Tris-EDTA buffer and a pressure cooker. Proteinase K was then incubated with the sample slices to facilitate proteinase digestion, exposing RNA targets of interest. After 5 min of neutral buffered formalin fixation, the FFPE slices were subjected to the GeoMx WTA panel (NanoString Technologies) overnight within a hybridization furnace, shielded from light. Fluorescently labeled markers (Pan-cytokeratin, CD45, and Syto13) were applied to the FFPE slices for morphology staining. UV laser dissociation techniques were employed to generate oligo barcodes from the selected ROIs. These barcodes were subsequently sequenced and analyzed in the experimental workflow. As not all areas of the entire tissue were ESPL regions, only tissues from the selected ROIs were sequenced. ROIs were chosen based on positive Pan-cytokeratin staining and H & E staining to confirm NE, LGIN, and HGIN regions of epithelium.

## Data processing and analysis

To account for system and experimental bias, as well as variation in ROI size, individual ROIs' Digital count conversion (DCC) files were normalized using ERCC RNA spike-in controls before downstream processing. This quality control step generated normalization positive factors from individual ROIs. The ROI inclusion criteria were limited to a minimum surface area of $1.6 \times 10^4\ \mu m^2$ for WTA and minimum nuclei counts of 200. ROIs with normalization positive factors higher than 3 or lower than 0.3 were excluded from downstream analysis. QC-qualified ROI count files were then normalized by the Q3 (3rd quartile of all selected targets). The normalized data were log-transformed with or without being median-centered before comparison and plotting. All data processing and analysis were performed using DSP analysis software (GeoMx NGS Pipeline Version 2.0.0.16, GeoMx DSP Control Center V.2.4.2.2) and R version 4.2.1 with relevant packages. Hierarchical clustering and correlation matrix were done with "pheatmap" package (Version 1.0.12). The principal component analysis (PCA) was conducted by "FactoMineR" (Version 2.8) and "factoextra" (Version 1.0.7) packages. For the differential expression analysis, a non-parametric Mann-Whitney U test with a significant cut-off p value of 0.05 was used. In some cases, due to the limited number of probes and samples, the p value was presented without adjustment. Other relevant plots were generated by "ggplot2" package (Version 3.4.2). For function and pathway annotation and enrichment analysis, differentially expressed genes (gene symbols) were processed by clusterProfiler[52] package (Version 4.6.2).

## Machine learning models construction

K-nearest neighbor (knn) is non-parametric and predicts the class of a test point based on the majority class of its k-nearest neighbors in the training set. Logistic Regression (logre) is parametric and models the probability of a binary response variable using a logistic/sigmoid function. Support Vector Machine (svm) is parametric and can perform classification and regression tasks by finding the best hyperplane to separate different classes in the feature space. Random Forest (rf) is an ensemble algorithm that combines multiple decision trees to enhance predictive accuracy and reduce overfitting. Neural Network Regression (nnet) is a family of parametric algorithms inspired by the human brain's neuron behavior. It can handle various tasks, including regression and classification. Naive Bayes (nbs) is a probabilistic algorithm suitable for classification tasks, determining class probabilities using Bayes' theorem based on conditional probabilities. Decision Trees (rpart) are non-parametric algorithms used for classification and regression tasks. They recursively divide the feature space into regions with reduced impurity.

These are the major algorithms used in machine learning. We adopted these seven machine learning models to evaluate the efficacy of the prediction model by distinguishing N, L, and M based on co-DEGs (mlr package, Version 2.19.1). Two significant metrics, logloss value, and multiclass AUC value, were used to assess the machine learning algorithms' performance. Logloss measures the precision of probability estimates, while multiclass AUC evaluates the algorithm's ability to differentiate between multiple classes, providing a comprehensive evaluation of overall performance.

## Immunohistochemical analysis

Five micrometers FFPE slices were deparaffinized and rehydrated. Then, slices were immersed in sodium citrate buffer (10 mmol/L, pH 6.0) and boiled 90 s for antigen retrieval. Blocked the sections with 3% $H_2O_2$ for 5 min. Wash the sections two times in phosphate-buffered saline (PBS) and nonspecific reactions were blocked by 10% goat serum for 1 h at room temperature. Then, tissue sections were incubated with the designated specific primary antibody overnight at 4 °C. On the second day, tissue sections were incubated with the secondary antibody for 30 min at room temperature after three times PBS washing. Then, the tissue sections were incubated with the avidin-biotin-peroxidase complex for 30 min. Followed by washing with PBS, the antigen-antibody binding was visualized with a DAB stain kit and counterstained with hematoxylin. Microscopic imaging was performed to capture images of the sample slices, and subsequent semi-automated counting of positive staining was conducted using digital image viewing software (version 12.3.2.8013, Aperio Image Scope, Leica, Germany)[53]. The immunohistochemical scoring system was used for statistical analysis of the immunohistochemistry results. The scoring system assigned points based on the intensity of staining: weak positive (1 point), positive (2 points), and strong positive (3 points). The immunohistochemical score (IHC score) was calculated using the formula: IHC score = weak positive × 1 + positive × 2 + strong positive × 3.

## Immunofluorescence analysis

The Immunofluorescence assay performed on paraffin-embedded sections commenced with a 2 h baking process at 65 °C, followed by 45 min of rehydration. Subsequently, antigen retrieval was conducted by immersing the slides in antigen retrieval solution and heating them using a pressure cooker until boiling for 90 s. Thereafter, slides were washed with distilled water for 5 min and blocked with 5% goat serum for 1 h. An appropriate dilution of CD68 and α-SMA was added, and slides were incubated overnight at 4 °C. On the second day, slides underwent two rounds of washes with PBS before adding the corresponding fluorescent secondary antibody (1:500). Incubation occurred in the dark at room temperature for 50 min, followed by two additional washes with PBS. Next, the DAPI staining solution (1:100) was applied, incubated in the dark at room temperature for 3-5 min, and washed twice with PBS. Finally, the slides were sealed with mounting reagents. Images were acquired through confocal microscopy. Mean fluorescence intensity (AU) was analyzed by Image J (version 1.53k) software.

## Cell culture

KYSE150 cell line was obtained from Cell Bank of Chinese Academy of Sciences (cat: CBTCCCAS, Shanghai, China). KYSE140, KYSE450, KYSE510, KYSE30, KYSE70, and KYSE410 were preserved and donated by Professor Ziming Dong (the Department of Pathophysiology, school of basic medical sciences of Zhengzhou University). These cell

lines were validated by STR analysis. Professor Enmin Li (Shantou University) donated the normal human esophagus immortalized epithelial cell (SHEE)[54]. Human esophageal squamous cell lines (KYSE140, KYSE150, KYSE450, KYSE510, KYSE30, KYSE70, KYSE410) were plated on plastic tissue culture dishes and cultured in RPMI-1640 medium (Biological Industries) containing 10% fetal calf serum (VivaCell, Shanghai, China) and 1% penicillin/streptomycin (100 U/ml) in an incubator at 37 °C with 5% $CO_2$. Human normal esophageal cell SHEE was cultured using a proprietary medium with a confidential composition.

## Plasmid construction
Candidate genes ORF expression plasmid was constructed by You Bio Biology Company (Changsha, China). Short hairpin RNAs (shRNA) are designed and subcloned into PLKO.1 vector. All the plasmids were used after being confirmed by sequencing. The primers are listed below:

sh*TAGLN2*: CCGGGAACGTGATCGGGTTACAGATCTCGAGATCTG TAACCCGATCACGTTCTTTTTG; AATTCAAAAAGAACGTGATCGGGTT ACAGATCTCGAGATCTGTAACCCGATCACGTTC. sh*CRNN*: CCGGGAG GAATCAGACAACAGAGATCTCGAGATCTCTGTTGTCTGATTCCTCTTT TTG; AATTCAAAAAGAGGAATCAGACAACAGAGATCTCGAGATCTCTG TTGTCTGATTCCTC.

## Cell transfection and lentiviral infection
The lentiviral shRNA/ORF plasmid was co-transfected with packaging vectors (pMD2G and psPAX2) into HEK293T cells using Simple-Fect transfection reagent (Signaling Dawn Biotech, Wuhan, China). After transfection, virus particles were harvested at 24 and 48 h and filtered using a 0.45 μm filter. To establish knockdown stable cell lines, cells were infected with the designated virus particles along with 8 μg/ml polybrene. After 24 h, the medium was changed, and the cells were selected with puromycin. Stable knockdown cell lines were generated through 2-3 passages. The knockdown and overexpression efficiency were evaluated by western blot.

## Cell proliferation assay and clonogenic formation assay
To assess the impact of candidate genes on cell survival, knockdown and overexpression cells were seeded in 96-well plates at a density of 2500 cells per well. Cell proliferation was measured using the MTT method, and absorbance was recorded after 0, 24, 48, and 72 h of incubation. For the evaluation of cell clonogenic formation ability, 250 viable cells were seeded in each well of a 6-well plate containing 2 ml of culture medium. After one or two weeks of incubation, the colonies were stained with 0.5% crystal violet, and the number of colonies was counted for further analysis.

## Anchorage-independent cell growth
The knockdown and overexpressed cells were mixed with 0.3% agar to form a top layer over a base layer of 0.5% agar. The plates were maintained at 37 °C in a 5% $CO_2$ incubator. The colonies were taken pictures and counted using Image J (version 1.53k).

## Organoid culture and analysis
Esophageal PDX-derived organoids (PDXO) were generated from esophageal patient-derived xenograft models LEG382 and LEG417 following the protocols outlined by Karakasheva et al.[55]. The dissociation procedure involves transferring and mincing tumor tissue followed by incubation with HBSS-DF, with collagenase IV (Thermo, 17104019) and Y-27632 (MCE, HY-10071) additives to enhance single-cell yields. Subsequently, the tissue fragments were filtered, and the collected cell suspension was centrifuged and resuspended. Cell density and viability were evaluated using the Trypan Blue exclusion test. For PDXO initiation, cells were seeded after the thawing of Matrigel and pre-warming of the plate and organoid medium. Organoid generation was carried out in a 6-well plate, where cells are transferred,

centrifuged, and resuspended in Matrigel before dispensed into the wells. After solidification, the organoid medium was added, and the organoids were cultivated for 10–14 days.

Lentivirus-mediated transduction of PDXO was conducted following the protocol outlined by Van Lidth de Jeude JF et al.[56]. The organoids were transferred into a 48-well plate with a small amount of medium. The designated lentivirus in transduction medium was added to the well, and to enhance transduction efficiency, spinoculation was performed by placing the plate containing organoids in a pre-warmed centrifuge at 32 °C and rotating it at 600 × g for 1 h. Afterward, the organoid-virus mixture was transferred to a culture incubator and incubated for 3 h at 37 °C to facilitate transduction. Next, the organoid-virus mixture was resuspended with culture medium and transferred to a microcentrifuge tube. The tube was then centrifuged at 850 × g for 5 min to pellet the organoids. The organoids were resuspended and seeded in a 96-well plate with Matrigel and medium. After 24 h, the medium was refreshed with 4 μg/ml puromycin and incubated for 48 h. Subsequently, the medium was changed without puromycin, and images were taken 10 days after transfection. Organoid numbers were calculated using Image J software (version 1.53k).

## PDX model establishment and treatment
Eight-week CB17 SCID female immunodeficient mice were obtained from Cyagen Biosciences, Inc. (Suzhou, Jiangsu, China). This animal study was approved by the Ethics Committee of China-US (Henan) Hormel Cancer Institute (Zhengzhou, Henan, China). The patients were consented for the generation of patient-derived xenograft and organoid models. The housing conditions for mice included a 12 h dark/light cycle, with an ambient temperature maintained at approximately 24 °C and a humidity level of around 60%. The PDX tumors were transplanted into mice and the experiment started on the 7th day after the transplant (showed as day 1). Tumor volume and mice body weight was measured twice a week and at the same time conducted intratumor injection of the designated virus (shNC, sh*TAGLN2*, NC, and *CRNN*). Tumor volume was calculated according to the following formula: length × width × height × 0.52. On day 46 after euthanizing the mice, tumors were removed, weighed, and photographed. The study adhered to the regulations of the ethics committee, and the maximum allowable tumor volume was set at 1000 mm$^3$. Notably, the tumor volumes observed in this study did not exceed the specified limit.

## RNA isolation and Reverse transcription polymerase chain reaction (RT-PCR) assay
Total RNA extraction was performed using the FastPure Cell/Tissue Total RNA Isolation Kit (Vazyme, RC101-01, Nanjing, China). Subsequently, cDNA synthesis was carried out utilizing the HiScript III 1st Strand cDNA Synthesis Kit (+gDNA wiper) (Vazyme, R312-01/02, Nanjing, China). The RT-PCR was conducted using the QuantiNovaTM SYBR Green PCR kit (cat. 208052, QIAGEN Sciences, Inc., Gaithersburg, MD, USA), and specific primers were utilized for the RT-PCR analysis. Signal analysis was performed using the Applied Biosystems 7500 FAST qPCR system (Thermo Fisher Scientific, Waltham, MA, USA). The expression of mRNAs was normalized to GAPDH. The primers used in this study: TAGLN2: TCCAGAACTGGCTCAAGGATGG; TCTGCTCCAT CTGCTTGAAGGC; KRT16: CTACCTGAGGAAGAACCACGAG; CTCGTAC TGGTCACGCATCTCA; KRT17: ATCCTGCTGGATGTGAAGACGC; TCCA CAATGGTACGCACCTGAC; CRNN: GGAGCTGAAAAGACTCTTGGAGC; CTGTGTGGTCTTCATCCAGCAG; MAL: CCATCACGATGCAAGACGGC TT; AGAACACCGCATGGACCACGTA; GAPDH: GTCTCCTCTGACTTCA ACAGCG; ACCACCCTGTTGCTGTAGCCAA.

## Western blotting
Cells were collected using a cell scraper on ice and treated with RIPA lysis buffer to extract cellular proteins. Protein quantification was performed, and the lysates were denatured with a sampling

buffer before being loaded onto an SDS-PAGE gel for electrophoresis. After separation, the proteins on the gel were transferred to a 0.45 µm PVDF membrane. The membrane was then incubated with specific antibodies to target the proteins of interest, and the bound antibodies were detected using chemiluminescence (ECL) reagent.

## Statistics and reproducibility
Statistical details of experiments and analyses can be found in the corresponding figure/ supplementary figure legends. In this study, each experiment was repeated at least three times. Data analysis was conducted using GraphPad Prism (Version 9.5.0) and R (version 4.2.1). $P$ value < 0.05 was considered a statistically significant difference. The sample size was not predetermined using any specific statistical method, and no data were excluded from the analyses.

## Illustrations
In the figures, the elements were created using BioRender (https://biorender.com/) and Procreate software.

## Reporting summary
Further information on research design is available in the Nature Portfolio Reporting Summary linked to this article.

## Data availability
The raw sequence data generated in this study have been deposited in the GSA-Human database under accession code HRA003627. The raw sequence data are available under restricted access for research purposes only, access can be obtained by the DAC (Data Access Committees) of the GSA-human database. According to the guidelines of GSA-human, all non-profit researchers are allowed access to the data, and the Principle Investigator of any research group is allowed to apply for Controlled-access of the data. The user can register the GSA database (https://ngdc.cncb.ac.cn/gsa-human/) and request the data. The approximate response time for accession requests is about 3 days. The access authority can be obtained for Research Use Only. The user can also contact the corresponding author directly. Once access has been approved, the data will be available to download for 2 months. The remaining data are available within the Article, Supplementary Information, or Source Data files. Source data are provided with this paper.

## Code availability
The codes for data analysis used in this study are available at https://github.com/hrcnlab/escc_pipline.

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

## Acknowledgements

The study was supported by The Affiliated Cancer Hospital of Zhengzhou University and the Tianjian Laboratory of Advanced Biomedical Sciences. This work was supported by the National Natural Science Foundation of China [No. 82073075 (Z.D.), 82172996 (X.Li), 82203290 (S.Z.), 81802875 (X.Liu), 81802876 (Y.G.)]; Major Science and Technology Projects in Henan Province (No. 221100310100, Z.D.); Training plan for young backbone teachers of Henan Province (No. 2020GGJS010, X.Li); Basic research and Cultivation Fund for young teachers of Zhengzhou University (No. JC202035023, X.Li); Science and technology innovation talents support plan of Henan Province (No.21HASTIT048, X.Li); The Science and Technology Project of Henan Province (No. 212102310880, 22102310416, S.Z.); The Key projects jointly built by Henan Province and the Ministry of Finance (No. SBGJ202102071, S.Z.); Natural Science Foundation of Henan (No.222300420390, L.Z.).

## Author contributions

Z.D., X.Li, S.Z., and X.Liu contributed to the conceptualization of the study. X.Liu, M.J., Y.G., R.Z., and L.Z. were involved in the development of the methodology. X.Liu, R.Y., L.Z., S.Y., W.Z., W.L., B.L., F.L., and Z.Z. carried out the investigation. K.W., X.Liu, R.Y., and L.Z. were responsible for data visualization. Z.D., X.Li, S.Z., and K.L. provided supervision. X.Liu wrote the original draft, and Z.D., K.L., X.Li, X.Liu, S.Z., M.J., and Y.G. contributed to the review and editing process.

## Competing interests

The authors declare no competing interests.
