## [Peer Review File · Nature Communications]

REVIEWER COMMENTS

Reviewer #1 (Remarks to the Author): Expert in oesophageal cancers, immunology, genomics, immunotherapy, and mouse models

In this study from Liu et al the transition of normal esophageal epithelium to precursors of esophageal squamous cell carcinoma (ESCC) such as low-grade (LGIN) and high-grade (HGIN) intraepithelial neoplasia is studied to find biomarkers to select patients with a high risk of cancer development. Using novel spatial whole transcriptome analysis, the team can study epithelial cells at different stages with unprecedented detail while studying the immune cells separately. To my knowledge this is the first study using spatial WTA to study ESCC development.

Thereby the team found that the pathogenesis of ESPL to ESCC is accompanied by the acquisition of more immune suppressive properties (more MDSCs, CAFs and protumoral cytokines). Furthermore, genes TAGLN2, KRT16, and CRNN seemed to be biomarkers for disease progression and these genes were further tested in a historical cohort and a cohort from which scRNAseq data was available and TCGA as well as organoid and PDX models.

The teams hypothesizes that when the major differences between normal-precursor-and cancer are identified that these differences can be used as biomarker for risk to develop cancer but this is not necessarily true. While this studies give a detailed description of different stages of ESCC development it is not clear how the differentially expressed genes can be used a biomarker. I therefor wonder whether the team should focus on the biology and not so much on biomarkers.

I appreciate how the study aims to make a translation towards the clinic, but clearly too limited detail is provided about the patient series. As not every LGIN/HGIN will progress to cancer it is important to know what kind of tissues are used of this study. The optimal situation is studying HGIN or LGIN in a patient that developed cancer later on, or HGIN adjacent to cancer but it is not clear what kind of tissues are used of this study which I consider a major limitation.

Comments and remarks:

1. Please provide more detail about the patient series in the methods: are all tissues from different patients or are there ESPL tissues and ESCC from the same patient? Please provide info about HGIN: did these patients develop cancer later?
2. Also: which AOIs were analyzed in the microenvironment analyses? Was this a stromal, non-CK compartment or CD45+ cells. For this part, information about the exact analysis is missing in the main document.
3. Furthermore: how many slides were used for this study? Were the NE, LGIN, HGIN and ESCC randomly distributed over the slides to prevent a slide effect?

4. Within the analyses of the microenvironment, the group analyzed 4 functional groups based on a previous publication while this was THE opportunity to study the main features of the microenvironment in a less biased way. Please explain the reason behind this and please analyze the data in a less biased fashion.
5. PCA analyses in Figure 3 does not indicate that HGIN shows more similarities with ESCC than LGIN. Please explain. Do these findings support the concept that ESCC always develop from HGIN..? Again: what is known from these cancer? Did any of these progress to cancer at a later time point?
6. The team tries to identify genes associated with risk but this information does not come from the genes but from the course of the disease....As ESCC showed also similarities to LGIN in the PCS, this study does not provide data to support the statement "Thus, most LGIN patients don't need extra treatment, just like clinical routine management seems reasonable".
7. Next the group performed pseudotime analysis by SCORPIUS to understand more about the developmental trajectories. Can this methods be used for this type of analyses? I only know it for scRNAseq data. Please consult a bioinformatician for advice.
8. The primary aim of the study is to find biomarkers. The experiments that are done to identify a role for some markers are described in very limited detail and put in the supplementary file. Please give a detailed description of this part of the study.
9. Page 8 : please do not use the term "normal patients" which implicates that the other patients are abnormal. Use patients with normal mucosa or patients with escc instead.
10. TAGLN2, KRT16, and CRNN were selected as biomarkers. Please describe the size of the series in which these markers were tested in LGIN and HGIN. For the patients with HGIN: did these develop into cancer?
11. The series for scRNA seq is not described.
12. It is not clear how bulk sequencing data from TCGA supports this story.
13. I do not agree that functional studies show evidence that genes can be used as biomarker.

Reviewer #2 (Remarks to the Author): Expert in ESCC in vivo and organoid models, ESCC precursors, and biomarkers

The study "Discovered biomarkers for predicting esophageal squamous precancerous lesions to cancer transition" by Liu et al utilizes spatial transcriptomics on human tissues across the pathogenesis of ESCC to identify TAGLN2, KRT16, and CRNN as factors that have potential to serve as biomarkers for ESCC

progression. These genes are then manipulated in vitro and in PDX tumors to investigate their effects on malignant properties of ESCC. Our understanding how ESCC develops from its premalignant precursor lesions remains quite limited and has great potential to guide clinical care in patients with ESCC precursor lesions. While the current study provides novel insights into ESCC progression, there are critical issues that must be addressed to clarify the experimental design/analysis as well as the biomarker potential of identified genes.

The methods for transcriptomics analysis and subsequent data processing and analysis are nearly absent from the methods section.

There is ambiguity in the sample size of each group with regard to number of independent tissue specimens analyzed as well as the number of ROIs analyzed in each tissue section. In line 481-482 it appears that the following number of independent tissue specimens (i.e. from individual human subjects) was used for each group: 6 NE, 6 LGIN, and 7 ESCC. It is then stated that "ESCC tissue samples belong to one TMA slide, and 1 ROI selected from one patient". Are ESCC samples from the same subjects from whom NE, LGIN, and HGIN were obtained? Additionally, the number of ROIs from groups other than ESCC is stated to be (N=11 ROIs), LGIN (N=12 ROIs) and HGIN (N=12 ROIs). It is unclear how 2-4 ROIs were selected for the 6 NE samples, yet the total number of ROIs is 11. The same is true for LGIN and HGIN. The number of ROIs in Figure S1 also does not seem to match with the total ROIs described above. A clear description of the methods for spatial transcriptomics, particularly as it relates to the human subjects and stages of ESCC, is critical.

Overall, Figure 1 is not well-integrated into the manuscript. Inclusion of panel A in Figure 1 is not helpful. It summarizes data from an independent study and terminology for lesions does not match that used in the current study. In my opinion the text describing these data in the introduction is sufficient. Validation of increased macrophage and fibroblast abundance is not provided. Statistical test and meaning of indicators of significance is not clear in panels B, E, F. Although there is a nice discussion of this current study as it relates to Yao et al. (PMID 32709844) in terms of findings in epithelium, inflammatory cells and fibroblasts are not discussed.

The authors note that they select DEGs that show continuous changes in expression at LGIN through ESCC as such genes may accelerate ESCC progression and, thus, serve as biomarkers. The experimental approach of inhibiting DEGs that are upregulated or overexpressing DEGs that are down regulated using ESCC cell lines provides evidence that TAGLN2, KRT16, and CRNN accelerate malignancy in ESCC cells, however, it is unclear that they accelerate the progression of ESCC.

Protein levels should be shown in cell lines along with a normal esophageal cell lines and also in overexpression and knockdown experiments.

There is literature related to TAGLN2, KRT16, and CRNN in various types of cancer, including ESCC, that is largely ignored.

Language in the manuscript should be edited for accuracy. For example, at several points the authors note a gradual progression in expression of a marker of interest in relation to stages of ESCC progression, yet data referenced fails to show a gradual progression (e.g. TAGLN2, CRNN in Fig 5C). Additionally, the authors should consider scaling back on the use of words like “tremendous” which may be interpreted as hyperbolic.

Reviewer #3 (Remarks to the Author): Expert in spatial transcriptomics, computational cancer genomics and machine learning

In this study, the authors searched for genes whose expressions are associated with distinct development stages, from precancer lesions to esophageal tumors. Their analysis identified TAGLN2, KRT16, and CRNN as top associates with esophageal tumor progression. Genetic knockdown of TAGLN2 and KRT16, and CRNN over-expression inhibited ESCC cell lines and PDX tumors. This study covers an essential and not intensively studied topic of cancer risk biomarkers from precancer lesions. However, I have a few comments and concerns that I hope the authors can address.

Point 1, More genetic experimental controls are needed to support the regulatory effects of TAGLN2, KRT16, and CRNN. Both shRNA and gene over-expression may lead to unexpected proliferation suppression due to off-target effects or unphysiologically high amplification of target proteins. The authors should demonstrate results (growth rates of cell line and tumor models) from rescuing TAGLN2 and KRT16 through over-expression and inhibiting CRNN through shRNA or CRISPR KO.

Point 2, Further interpretation of associations with immune cell types (lines 297, 298, Supplementary Figure 5F). The manuscript only briefly mentioned these results as “these genes correlated with at least one kind of immune cells.” However, it will be more informative to list the main manuscript of the specific immune cell types and the biological interpretations.

Point 3, Please describe the machine learning and prediction models in the main text (lines 234 - 236). Currently, there is only one sentence in the main manuscript. Readers may not always look into the method section. So, more details are always more helpful.

For example, is the prediction model trying to predict N, L, and M states from the gene expression of a sample itself? Or is the prediction model trying to predict whether the sample will progress into L or M

using only gene expression from “N” sample? Also, why did the logistic regression and KNN perform better than other models?

Responses to Reviewer #1:

Reviewer #1 (Remarks to the Author): Expert in oesophageal cancers, immunology, genomics, immunotherapy, and mouse models

In this study from Liu et al the transition of normal esophageal epithelium to precursors of esophageal squamous cell carcinoma (ESCC) such as low-grade (LGIN) and high-grade (HGIN) intraepithelial neoplasia is studied to find biomarkers to select patients with a high risk of cancer development. Using novel spatial whole transcriptome analysis, the team can study epithelial cells at different stages with unprecedented detail while studying the immune cells separately. To my knowledge this is the first study using spatial WTA to study ESCC development.

Thereby the team found that the pathogenesis of ESPL to ESCC is accompanied by the acquisition of more immune suppressive properties (more MDSCs, CAFs and

protumoral cytokines). Furthermore, genes TAGLN2, KRT16, and CRNN seemed to be biomarkers for disease progression and these genes were further tested in a historical cohort and a cohort from which scRNAseq data was available and TCGA as well as organoid and PDX models.

Q1: The teams hypothesizes that when the major differences between normal-precursor-and cancer are identified that these differences can be used as biomarker for risk to develop cancer but this is not necessarily true. While this studies give a detailed description of different stages of ESCC development it is not clear how the differentially expressed genes can be used a biomarker. I therefor wonder whether the team should focus on the biology and not so much on biomarkers.

A1: Thank you for your suggestions. We agree with your idea that we should focus on biology and not so much on biomarkers, due to the limited number of tested samples. Given the limited sample size at our study, referring to the genes as biomarkers was deemed inappropriate. In order to improve the precision of the language used in the manuscript, we have replaced the “biomarker” with “candidate indicator”. The primary focus of this study is to ascertain whether our candidate genes are associated with the progression of ESCC. As we validated TAGLN2 and CRNN correlated with ESCC progression, from these results, TAGLN2 and CRNN may serve as candidate indicators of the risk of developing ESCC.

Q2: I appreciate how the study aims to make a translation towards the clinic, but clearly too limited detail is provided about the patient series. As not every LGIN/HGIN will progress to cancer it is important to know what kind of tissues are used of this study. The optimal situation is studying HGIN or LGIN in a patient that developed cancer later on, or HGIN adjacent to cancer but it is not clear what kind of tissues are used of this study which I consider a major limitation.

A2: The samples information was added in the Methods section of the revised manuscript. Yes, optimal situation is studying HGIN or LGIN in a patient that developed cancer later on, but we cannot obtain such kind of samples when this project first started. And the transcriptome change during ESCC tumorigenesis is not well

studied before, thus we first want to see what happens during ESCC tumorigenesis. Through this process, we identified genes that were altered in association with the development of ESCC, suggesting their potential relevance to the occurrence and progression of ESCC. Thus, we started to investigate whether our candidate genes are associated with the progression of ESCC.

We opted not to select surgical samples from ESCC patients that contained LGIN and HGIN adjacent to the cancerous area. This decision was based on the fact that the patients had already been diagnosed with ESCC. Consequently, it is unclear whether there are alterations in the gene or transcriptome information of the LGIN and HGIN adjacent to the cancerous area when compared to patients with LGIN and HGIN under normal conditions without a history of ESCC.

In this study, LGIN, HGIN, and ESCC samples were collected from different patients to investigate transcriptome change during ESCC tumorigenesis. The HGIN samples were obtained from endoscopic submucosal dissection (ESD) samples, from which we cannot get more information about whether it developed or not later on for the removal of ESPL lesion.

To solve the problem whether our candidate genes are associated with the progression of ESCC, we finally collected paired tissue samples obtained from initial diagnoses and samples obtained at later progression stages of the same patient for further validations in the revised manuscript.

Q3: Please provide more detail about the patient series in the methods: are all tissues from different patients or are there ESPL tissues and ESCC from the same patient? Please provide info about HGIN: did these patients develop cancer later?

A3: In this study LGIN, HGIN, and ESCC samples were collected from different patients.

Up until now, the HGIN patients in question have not developed cancer. This is primarily because the HGIN samples were obtained from endoscopic submucosal dissection (ESD) samples after the lesions had been removed. To explore the possibility

of a correlation between the indicators discovered in this study and ESCC progression, the revised manuscript comprises paired tissue samples obtained from initial diagnoses and paired tissue samples obtained from initial diagnoses and of the same patient. This was done to establish a clear relationship. Additional details are provided to answer your question 5 (A5), a similar question with this one.

Detailed information on the sample size, distribution across slides, and number of ROIs analyzed is presented below. These additional details are comprehensively documented in the Materials and Methods section. (page 21, line1187-1209).

The tissue samples utilized for spatial WTA sequencing were obtained from patients diagnosed with LGIN, HGIN, or ESCC (N=6, 6, and 7, respectively). Specifically, the NE group samples were selected from the normal epithelial region of LGIN patients, indicating that NE and LGIN samples originate from the same person (N=6). To ensure experimental accuracy, a total of four slides were employed for spatial WTA sequencing. Slides 1-3 contained ESPL tissues, each slide consisting of two LGIN and two HGIN patient tissues. Slide-4 comprised an ESCC tissue microarray (TMA), with each spot corresponding to one ESCC patient sample. Each ESPL slide included different sample types (NE, LGIN, and HGIN), which effectively minimized potential slide effects. Given the larger area of ESPL samples, each LGIN sample was subjected to the selection of 3 to 4 ROIs, including 2 ROIs from the LGIN region and 1 to 2 ROIs from the NE region. Each HGIN sample was subjected to the selection of 2 ROIs from the HGIN region. In contrast, due to the smaller tissue area of ESCC tissue samples, only one ROI was selected per patient sample. Ultimately, a total of 11 ROIs for NE, 12 ROIs for LGIN, 12 ROIs for HGIN, and 7 ROIs for ESCC were collected and sequenced using spatial whole-transcriptome profiling. All sequenced ROIs' data were incorporated into further analysis.

A4. Also: which AOIs were analyzed in the microenvironment analyses? Was this a stromal, non-CK compartment or CD45+ cells. For this part, information about the exact analysis is missing in the main document.

A4: Microenvironment analysis in this study was carried out based on all the ROIs' spatial WTA sequencing results. Since not all areas of the entire tissue are ESPL region, only tissue from these selected ROIs were sequenced. ROIs were chosen based on positive pan-CK staining and HE staining confirmed NE, LGIN, HGIN regions of epithelium. Therefore, all analyses, including microenvironment analysis in this paper, were conducted based on data from these ROIs. This information has been added to the Materials and Methods section (page 22, line 1229-1232). In fact, we intended to describe the state of the ESPL stage in the microenvironment section. However, the term 'microenvironment' is not accurate. Therefore, we decided to change the title of this part to ESPL status analysis. Based on your suggestion to use unbiased analytical methods, we reanalyzed this part and elaborated on the status of ESPL stage from aspects such as abnormal metabolism, abnormal signal transduction pathways, immune microenvironment, and disease-/cancer-related pathways.

Q5. Furthermore: how many slides were used for this study? Were the NE, LGIN, HGIN and ESCC randomly distributed over the slides to prevent a slide effect?

A5: A total of four slides were employed for spatial WTA sequencing. Slides 1-3 contained ESPL tissues, each slide consisting of two LGIN and two HGIN patient tissues. Slide-4 comprised an ESCC tissue microarray (TMA), with each spot corresponding to one ESCC patient sample. Each ESPL slide included different sample types (NE, LGIN, and HGIN), which effectively minimized potential slide effects. For spatial WTA-related steps, all slides were operated at the same time. These details have been added to the Materials and Methods section (page 21, line 1197-1201). The distribution of samples is shown below.

Q6. Within the analyses of the microenvironment, the group analyzed 4 functional groups based on a previous publication while this was the opportunity to study the main features of the microenvironment in a less biased way. Please explain the reason behind this and please analyze the data in a less biased fashion.

A6: Thanks for your suggestion, we decided to analyze the data in a less biased way rather than analyze the data based on a publication. We reanalyzed this part and elaborated on the status of ESPL stage from aspects such as biological processes that occur during tumorigenesis, abnormal metabolic pathways that occur in tumors, abnormal signal transduction pathways in cancer, and immune-related pathways (page 5-6). The details showed in the results section with the title ESPL status analysis. Here is the figure that was previously mentioned:

Q7. PCA analyses in Figure 3 does not indicate that HGIN shows more similarities with ESCC than LGIN. Please explain. Do these findings support the concept that ESCC always develop from HGIN..? Again: what is known from these cancer? Did any of these progress to cancer at a later time point?

A7: Upon consulting a bioinformatician, it was determined that the previous PCA figure displayed in Figure 3 was not accurate. As such, a new figure was created and is shown below. The updated figure demonstrates that both NE and ESCC samples could be clearly differentiated from one another. Additionally, LGIN showed greater similarity to NE than HGIN did, and conversely, HGIN displayed more similarity to ESCC than LGIN did. The proximity of some LGIN samples to HGIN indicated the likelihood of progression from LGIN to HGIN stage. These findings support the notion that HGIN has a greater risk of developing into ESCC compared to LGIN.

Due to the desire to obtain larger tissue samples, most samples were acquired through endoscopic submucosal dissection (ESD), which made it difficult to determine whether these patients had progressed after the lesion was removed.

To investigate these candidate indicators associated with progression, paired tissue samples were obtained from patients who were initially diagnosed with LGIN and later progressed to HGIN, or those who were first diagnosed with HGIN and later progressed to ESCC for IHC validation. Due to the rarity of paired tissue samples from the same individual corresponding to before- and after-progression, as well as the typically several years required for LGIN progression to ESCC, paired tissue samples from individuals who progressed from LGIN to ESCC were unavailable. The tissue taken during the initial diagnosis was labeled as “before-progression” (Before) and the tissue procured from later diagnoses were designated as “after-progression” (After). Analysis of the IHC scores of all these paired progressed samples revealed that TAGLN2 expression significantly increased in the after-progression group tissues compared with before-progression group, while CRNN expression decreased. These results indicated that TAGLN2 and CRNN are correlated with ESCC progression. In contrast, KRT16 did not exhibit any significant changes in these paired samples, indicating that it cannot be used as a candidate indicator of ESCC progression. Therefore, KRT16 was excluded, and only TAGLN2 and CRNN were included in the final manuscript.

Q8. The team tries to identify genes associated with risk but this information does not come from the genes but from the course of the disease....As ESCC showed also similarities to LGIN in the PCA, this study does not provide data to support the statement “Thus, most LGIN patients don’t need extra treatment, just like clinical routine management seems reasonable”.

A8: As you mentioned “The team tries to identify genes associated with risk but this information does not come from the genes but from the course of the disease”, the finding that LGIN can progress to HGIN, which may further progress to ESCC, and even HGIN may develop into ESCC in clinical practice, it suggesting that certain genes may be associated with an increased risk of progression. Consequently, we undertook efforts to identify such genes.

From the updated PCA figure, LGIN showed greater similarity to NE than HGIN did, and conversely, HGIN displayed more similarity to ESCC than LGIN did. The presence of some LGIN samples in close proximity to HGIN samples suggests that some LGIN tissues share a similar transcriptional background with HGIN, which may indicate the possibility of progression from LGIN stage to HGIN stage, and eventually ESCC. The best way is to predict the progression risk of patients and give treatment to these high-risk patients. In this paper, we aimed to identify candidate indicators associated with disease progression. According to your question, our study does not provide data to support the statement “Thus, most LGIN patients don’t need extra treatment, just like clinical routine management seems reasonable”, we agreed with your idea, and we decide delete this statement in the manuscript.

Q9. Next the group performed pseudotime analysis by SCORPIUS to understand more about the developmental trajectories. Can this method be used for this type of analyses? I only know it for scRNAseq data. Please consult a bioinformatician for advice.

A9: Yes, this method can be used for this type of analyses. Although pseudotime analysis is a computational method commonly applied in the analysis of single-cell gene expression data to infer developmental trajectories, cellular differentiation processes, cell cycle progression, and other temporal dynamics. After consulting with a bioinformatician, we discovered references that indicate the viability of utilizing pseudotemporal trajectories analysis for sequencing data other than scRNAseq. Please find references to these sources listed below. We also added these references to the manuscript.

Campbell KR, Yau C. Uncovering pseudotemporal trajectories with covariates from single cell and bulk expression data. *Nat Commun.* 2018 Jun 22;9(1):2442.

Li B, Cui Y, Nambiar DK, Sunwoo JB, Li R. The Immune Subtypes and Landscape of Squamous Cell Carcinoma. *Clin Cancer Res.* 2019 Jun 15;25(12):3528-3537.

Saelens W, Cannoodt R, Todorov H, Saeys Y. A comparison of single-cell trajectory inference methods. *Nat Biotechnol.* 2019 May;37(5):547-554.

Q10. The primary aim of the study is to find biomarkers. The experiments that are done to identify a role for some markers are described in very limited detail and put in the supplementary file. Please give a detailed description of this part of the study.

A10: Thanks for your suggestions. In response to the feedback we received, we have made significant revisions to this section by including a more detailed explanation of our machine learning techniques, the criteria employed for candidate gene selection, and the gradual process of narrowing down these candidates to arrive at the final candidate indicators. We have also made adjustments to the figure presented in this section. As we have combined the results obtained from the knn and logre algorithms into one figure (Figure 5b), thus, we have included the separate outcomes in the supplementary file. Additionally, we have moved the figure ROC plots of GEO data set and overall performance metrics figure for N, L, and M groups from the supplementary figures to the main section figures. During the revision process, we reanalyzed the immunohistochemical results using an immunohistochemical scoring system method to better align our approach with clinical diagnosis. As a result, we updated figures presented in relevant sections of the manuscript.

The candidate indicator discovery steps including: (1) Get candidate indicators list. The DEGs started from LGIN stage and continuous changing during all the stages may accelerate ESCC tumorigenesis. To find some potential indicators that indicating the risk of developing ESCC when patients are diagnosed with ESPL, we analyzed the co-DEGs expression level at NE, LGIN, HGIN and ESCC stage. (2) Narrow down the list

of candidate genes by machine learning method. First, selected optimal algorithms from seven major algorithms. After thorough evaluation, we identified two algorithms knn and logre that displayed superior performance than other algorithms across both metrics, characterized by a smaller average logloss value and a higher AUC value. Second, obtained gene panel from knn and logre algorithms. Third, overlap the gene panel generated by the knn algorithm and the importance rank >50 genes acquired from the logre algorithm. The result revealed that CRNN, KRT17, MAL, KRT16, and TAGLN2 genes exhibited more significant importance than other genes. (3) IHC staining for further evaluate the possibility of candidate genes as ESPL and ESCC indicator. The gene list narrowed down to TAGLN2, CRNN, KRT16 after IHC screening step. (4) To investigate candidate indicators associated with progression, paired tissue samples were obtained from patients who were initially diagnosed with LGIN and later progressed to HGIN, or those who were first diagnosed with HGIN and later progressed to ESCC for IHC validation. After this step, KRT16 are excluded. Finally, TAGLN2 and CRNN are recognized as correlated to ESCC progression.

Q11. Page 8: please do not use the term “normal patients” which implicates that the other patients are abnormal. Use patients with normal mucosa or patients with escc instead.

A11: Thanks for your carefully reading and giving good suggestions, we instead “normal patients” by patients with normal mucosa in the revised manuscript.

Q12. TAGLN2, KRT16, and CRNN were selected as biomarkers. Please describe the size of the series in which these markers were tested in LGIN and HGIN. For the patients with HGIN: did these develop into cancer?

A12: TAGLN2, KRT16, and CRNN were tested in LGIN (N=19) and HGIN (N=19) samples by IHC staining. These results and the size of the series are shown in Figure 5 E-F. As we were unable to obtain any follow-up information regarding the samples collected, it remains unclear whether or not these patients had progressed at a later time.

To explore the possibility of a correlation between the candidate indicators discovered in this study and ESCC progression, this revised manuscript comprises paired tissue samples obtained from initial diagnoses and samples obtained at later progression stages of the same patient for validation.

Q13. The series for scRNA seq is not described.

A13: Thanks for your suggestions. We added more information of scRNA seq in the revised manuscript (page 14, line 607-616).

Q14. It is not clear how bulk sequencing data from TCGA supports this story.

A14: In this study, bulk sequencing data from TCGA as shown in Supplementary Figure 5b. mRNA expression of TAGLN2, KRT16, KRT17 in TCGA was high, while CRNN and MAL was low in ESCC compared with normal, which were consistent with the trend of our sequencing results. These data provide further evidence to support the credibility and accuracy of our sequencing results. The results of TCGA survival curves demonstrate a significant association between TAGLN2, KRT16, and CRNN and the prognosis of esophageal cancer. This suggests that these genes may be involved in the progression of ESCC, underscoring their potential value as indicators not only for disease progression but also for prognosis.

Q15. I do not agree that functional studies show evidence that genes can be used as biomarker.

A15: Yes, these functional studies do not prove that these genes can serve as candidate indicators, they provide preliminary insights into the mechanisms by which these genes may contribute to the progression of ESCC. Specifically, TAGLN2 appears to promote tumor proliferation, while CRNN inhibits the growth of cancer cells. Normally, human normal esophageal cell line (SHEE) is unable to form colonies in anchorage-independent cell growth assays, but when the cells are stimulated with EGF, normal

cells tend to show signs of transformation into malignant cells. In normal esophageal cell, TAGLN2 promotes the transformation of normal cells to malignant cells, while CRNN inhibits this transformation. These findings suggest that TAGLN2 and CRNN may facilitate ESCC progression by regulating cell proliferation.

Reviewer #2 (Remarks to the Author): Expert in ESCC in vivo and organoid models, ESCC precursors, and biomarkers

The study “Discovered biomarkers for predicting esophageal squamous precancerous lesions to cancer transition” by Liu et al utilizes spatial transcriptomics on human tissues across the pathogenesis of ESCC to identify TAGLN2, KRT16, and CRNN as factors that have potential to serve as biomarkers for ESCC progression. These genes are then manipulated in vitro and in PDX tumors to investigate their effects on malignant properties of ESCC. Our understanding how ESCC develops from its premalignant precursor lesions remains quite limited and has great potential to guide clinical care in patients with ESCC precursor lesions. While the current study provides novel insights into ESCC progression, there are critical issues that must be addressed to clarify the experimental design/analysis as well as the biomarker potential of identified genes.

Responses to Reviewer #2:

Q1: The methods for transcriptomics analysis and subsequent data processing and analysis are nearly absent from the methods section.

A1: Thank you for your suggestion. We have incorporated the subsequent data processing and analysis methods into our Materials and Methods section with the title Data processing and analysis (page 22, line 1229-1248). In summary, the information presented below.

To adjust system and experimental bias and to counteract ROI size variation effects, Digital count conversion (DCC) files for individual ROIs were normalized by ERCC RNA spike-in controls before downstream processing. This quality control step generated normalization positive factors from individual ROIs. The ROI inclusion criteria were limited on a minimum surface area of $1.6 \times 10^4 \mu\text{m}^2$ for WTA, and minimum nuclei counts of 200. Any ROIs resulting in a normalization positive factor higher than 3 or lower than 0.3 were excluded from the downstream analysis. QC-qualified ROI count files were then normalized by the Q3 (3rd quartile of all selected

targets). The normalized data were log-transformed with or without being median-centered before comparison and plotting. All data were processed and analyzed in DSP analysis software and R version 4.2.1 with relevant packages. The correlation analysis was computed using the “Pearson” method. Hierarchical clustering and correlation matrix were done with “pheatmap” package. The principal component analysis (PCA) was conducted by “FactoMineR” and “factoextra” packages. Volcano plots were created with log₂FC set at 1 and adjusted p-value at 0.05 for cut-off (dashed lines). Venn plots were created with the “VennDiagram” package. For the differential expression analysis, a non-parametric Mann–Whitney U-test was used, and p-value was set to 0.05 at a significant cut-off. Due to the limited number of probes and samples, in some cases, p-value was presented without adjustment. Other relevant plots were generated by “ggplot2” package. For function and pathway annotation and enrichment analysis, differentially expressed genes (gene symbol) were processed by clusterProfiler package.

Q2: There is ambiguity in the sample size of each group with regard to number of independent tissue specimens analyzed as well as the number of ROIs analyzed in each tissue section. In line 481-482 it appears that the following number of independent tissue specimens (i.e. from individual human subjects) was used for each group: 6 NE, 6 LGIN, and 7 ESCC. It is then stated that “ESCC tissue samples belong to one TMA slide, and 1 ROI selected from one patient”. Are ESCC samples from the sample subjects from whom NE, LGIN, and HGIN were obtained? Additionally, the number of ROIs from groups other than ESCC is stated to be (N=11 ROIs), LGIN (N=12 ROIs) and HGIN (N=12 ROIs). It is unclear how 2-4 ROIs were selected for the 6 NE samples, yet the total number of ROIs is 11. The same is true for LGIN and HGIN. The number of ROIs in Figure S1 also does not seem to match with the total ROIs described above. A clear description of the methods for spatial transcriptomics, particularly as it relates to the human subjects and stages of ESCC, is critical.

A2: We apologize for not describing the sample information clearly and leaving the reader confused. The tissue samples utilized for spatial WTA sequencing were obtained from patients diagnosed with LGIN, HGIN, or ESCC (N=6, 6, and 7, respectively). Specifically, the NE group samples were selected from the normal epithelial region of LGIN patients, indicating that NE and LGIN samples originate from the same person (N=6). To ensure experimental accuracy, a total of four slides were employed for spatial WTA sequencing. Slides 1-3 contained ESPL tissues, each slide consisting of two LGIN and two HGIN patient tissues. Slide-4 comprised an ESCC tissue microarray (TMA), with each spot corresponding to one ESCC patient sample. Each ESPL slide included different sample types (NE, LGIN, and HGIN), which effectively minimized potential slide effects. Given the larger area of ESPL samples, each LGIN sample was subjected to the selection of 3 to 4 ROIs, including 2 ROIs from the LGIN region and 1 to 2 ROIs from the NE region. Each HGIN sample was subjected to the selection of 2 ROIs from the HGIN region. In contrast, due to the smaller tissue area of ESCC tissue samples, only one ROI was selected per patient sample. Ultimately, a total of 11 ROIs for NE, 12 ROIs for LGIN, 12 ROIs for HGIN, and 7 ROIs for ESCC were collected and sequenced using spatial whole-transcriptome profiling. All sequenced ROIs' data were incorporated into further analysis. These additional details have been documented in the Materials and Methods section (page 21, line 1183-1202).

Q3: Overall, Figure 1 is not well-integrated into the manuscript. Inclusion of panel A in Figure 1 is not helpful. It summarizes data from an independent study and terminology for lesions does not match that used in the current study. In my opinion the text describing these data in the introduction is sufficient.

A3: In response to your suggestion, we have removed Figure 1A and instead incorporated the associated data into our introduction section. Nevertheless, we chose to retain the remaining panels within Figure 1 as they outline the detailed procedure of spatial WTA analysis and demonstrate the pathological phenotypes of the samples,

while also providing an intuitive depiction of the ROIs utilized in WTA sequencing. These panels have been integrated into our main figure for the sake of clarity and brevity.

Q4: Validation of increased macrophage and fibroblast abundance is not provided.

A4: We have included additional validation experiments to detect the expression of macrophage and fibroblast markers (CD68 and α -SMA, respectively) in NE, LGIN, HGIN, and ESCC using immunofluorescence techniques. Based on our results, we observed a notable increase in CD68 and α -SMA expression levels in samples obtained from different disease stages (NE to ESCC), indicative of heightened abundance of macrophages and fibroblasts in these samples. The relevant experimental methods and results have been incorporated into the corresponding section of the article.

Q5: Statistical test and meaning of indicators of significance is not clear in panels B, E, F.

A5: Following consultation with a statistical expert, we reanalyzed the statistical results of fibroblasts and macrophages using the Brown-Forsythe and Welch ANOVA tests provided by Graphpad Prism for the comparison among groups. The statistical method added to the figure legend of Figure 2. The figure presenting the updated statistical analysis is located below this sentence. As suggested by other reviewers, we conducted a new analysis of the microenvironment section, leading to the exclusion of the original Figure 2B. These changes have been displayed in Figure 2.

Q6: Although there is a nice discussion of this current study as it relates to Yao et al. (PMID 32709844) in terms of findings in epithelium, inflammatory cells and fibroblasts are not discussed.

A6: Thank you for your feedback. We added more discussion about fibroblasts and inflammatory cells compared with Yao's study to the discussion section (page 19-20, line 1022-1037). The details showed below:

In the study of Yao et al., they found a fibroblasts cluster (FibC8) that increased along ESCC tumorigenesis with the high expression of genes from Myc and angiogenesis pathways. They showed the highest proportion of FibC8 cluster cells at ESCC stage. Similar with their study, we find angiogenesis related genes enriched in ESCC stage. In their study, a decrease was observed in the proportion of CD8+memory T cells as the tumorigenic process progressed beyond stage INF. This finding suggests that non-effective CD8+ T cell dominant microenvironment throughout the precancerous stages. In our study, we find similar decrease tendency of CD8+ memory T cells at ESPL stages (Figure 2). To some extent, the mice model results are consistent with human, thus, the mouse model induced by 4NQO serves as a suitable means of simulating the progression of esophageal cancer. It can offer critical insights for preclinical research into the process of esophageal precancerous lesions.

Q7: The authors note that they select DEGs that show continuous changes in expression at LGIN through ESCC as such genes may accelerate ESCC progression and, thus, serve as biomarkers. The experimental approach of inhibiting DEGs that are upregulated or overexpressing DEGs that are down regulated using ESCC cell lines provides evidence that TAGLN2, KRT16, and CRNN accelerate malignancy in ESCC cells, however, it is unclear that they accelerate the progression of ESCC.

A7: Yes, we agree with your idea that inhibiting DEGs that are upregulated or overexpressing DEGs that are down regulated using ESCC cell lines only provides evidence these genes accelerate malignancy in ESCC cells and cannot say they accelerate the progression of ESCC. We added additional experiments to verify whether

these candidates associated with progression including test these gene use paired before- and after-progression tissue samples from the same individual and the experiment validated the role of these genes in the transformation from normal esophageal cells to malignant cells. The details showed below:

To investigate these candidate indicators associated with progression, paired tissue samples were obtained from patients who were initially diagnosed with LGIN and later progressed to HGIN, or those who were first diagnosed with HGIN and later progressed to ESCC for IHC validation. Due to the rarity of paired tissue samples from the same individual corresponding to before- and after-progression, as well as the typically several years required for LGIN progression to ESCC, paired tissue samples from individuals who progressed from LGIN to ESCC were unavailable. The tissue taken during the initial diagnosis was labeled as “before-progression” (Before) and the tissue procured from later diagnoses were designated as “after-progression” (After). Analysis of the IHC scores of all these paired progressed samples revealed that TAGLN2 expression significantly increased in the after-progression group tissues compared with before-progression group, while CRNN expression decreased. These results indicated that TAGLN2 and CRNN are correlated with ESCC progression. In contrast, KRT16 did not exhibit any significant changes in these paired samples, indicating that it cannot be used as a candidate indicator of ESCC progression. Therefore, KRT16 was excluded, and only TAGLN2 and CRNN were included in the final manuscript.

In addition, we conducted experiments using human normal esophageal cell line (SHEE) to investigate the role of these genes in the transformation from normal cells to malignant cells. Specifically, we examined the overexpression of TAGLN2, which is low expressed in SHEE, and the knockdown of CRNN, which is high expressed in SHEE. Overexpression and knockdown efficiency of TAGLN2 and CRNN were detected by western blot (Figure 9a). Normally, SHEE cells are unable to form colonies in anchorage-independent cell growth assays, but when the cells are stimulated with EGF, normal cells tend to show signs of transformation into malignant cells, as indicated by the formation of clones (Figure 9b). By adding EGF to simulate the process

of normal cells transforming to malignant cells, we evaluated whether the genes accelerate the progression of ESCC. Results from cell proliferation experiments showed that overexpression of TAGLN2 or knockdown of CRNN promoted SHEE cell proliferation (Figure 9c). In anchorage-independent cell growth assays, TAGLN2 overexpression or CRNN knockdown significantly increased the colonies number compared to the control group, indicating that TAGLN2 promotes the transformation of normal cells to malignant cells, while CRNN inhibits this transformation (Figure 9d-e). Taking together the results of IHC analysis of paired samples before- and after-progression and normal esophageal cell transformation assay, it suggested that TAGLN2 and CRNN are associated with the progression of ESCC. Additionally, the mechanism of TAGLN2 and CRNN may facilitate ESCC progression by regulating cell proliferation. These experiments and results have been added to the revised manuscript.

Q8: Protein levels should be shown in cell lines along with a normal esophageal cell lines and also in overexpression and knockdown experiments.

A8: Thank you for your suggestions. Protein levels in cell lines along with a normal esophageal cell line and also in overexpression and knockdown experiments are prepared and added to Figure 8a, Supplementary Figure 8a. Due to the less correlation between KRT16, KRT17 and MAL and ESCC progression risk, we decided delete these genes' related data and only keep TAGLN2 and CRNN data in the revised manuscript.

Q9: There is literature related to TAGLN2, KRT16, and CRNN in various types of cancer, including ESCC, that is largely ignored.

A9: Thank you for your feedback. The functions of TAGLN2 and CRNN in different types of cancer have been elucidated in various literature. We have included relevant literature on these genes in the revised manuscript. Although, TAGLN2 is reported about the function in promoting cell proliferation, invasion, migration in some cancer

types such as colorectal cancer, gliomas and so on. The publications related to esophageal cancer usually reported TAGLN2 as a down-stream of microRNAs, but its role in esophageal cancer is not well studied. Thus, we still keep the functional study of these genes in this manuscript, just want to provide preliminary insights into the mechanisms of these genes in accelerating the progression of ESCC.

Q10: Language in the manuscript should be edited for accuracy. For example, at several points the authors note a gradual progression in expression of a marker of interest in relation to stages of ESCC progression, yet data referenced fails to show a gradual progression (e.g. TAGLN2, CRNN in Fig 5C). Additionally, the authors should consider scaling back on the use of words like “tremendous” which may be interpreted as hyperbolic.

A10: We greatly appreciate your suggestions. We agree with your feedback and have made revisions throughout the entire manuscript to ensure accuracy and precision in our language usage. In addition, given the limited sample size at our study, referring to the genes as biomarkers was deemed inappropriate. In order to improve the precision of the language used in the manuscript, we have replaced the “biomarker” with “candidate indicator”.

Reviewer #3 (Remarks to the Author): Expert in spatial transcriptomics, computational cancer genomics and machine learning

In this study, the authors searched for genes whose expressions are associated with distinct development stages, from precancer lesions to esophageal tumors. Their analysis identified TAGLN2, KRT16, and CRNN as top associates with esophageal tumor progression. Genetic knockdown of TAGLN2 and KRT16, and CRNN over-expression inhibited ESCC cell lines and PDX tumors. This study covers an essential and not intensively studied topic of cancer risk biomarkers from precancer lesions. However, I have a few comments and concerns that I hope the authors can address.

Responses to Reviewer #3:

Q1: More genetic experimental controls are needed to support the regulatory effects of TAGLN2, KRT16, and CRNN. Both shRNA and gene over-expression may lead to unexpected proliferation suppression due to off-target effects or unphysiologically high amplification of target proteins.

The authors should demonstrate results (growth rates of cell line and tumor models) from rescuing TAGLN2 and KRT16 through over-expression and inhibiting CRNN through shRNA or CRISPR KO.

A1: Thanks for your suggestions. We added experiment of cell proliferation, anchorage-independent cell growth assay and clonogenic formation assay from rescuing TAGLN2 through over-expression and inhibiting CRNN through shRNA. In the rescue group, we observed increased cell proliferation and colony formation via cell proliferation assay, anchorage-independent cell growth assay and clonogenic formation assay upon knocking down TAGLN2 and overexpressing it, or overexpressing CRNN and subsequently knocking it down (Supplementary Figure 8a-f). Given the significant concerns raised by other reviewers regarding the relationship between our candidate indicators and the progression risk of ESCC, this revision manuscript involved paired initial diagnoses and post-progression tissue samples. KRT16 did not exhibit significant changes in these paired samples, indicating that it cannot be used as a candidate indicator of ESCC progression. Therefore, KRT16 was removed from the main text,

and only TAGLN2 and CRNN were included in the final manuscript. To ensure that our article adheres closely to its main focus, we have made adjustments to the content and excluded phenotypic data related to KRT16, KRT17, and MAL genes that are not directly relevant to the topic.

Q2: Further interpretation of associations with immune cell types (lines 297, 298, Supplementary Figure 5F). The manuscript only briefly mentioned these results as “these genes correlated with at least one kind of immune cells.” However, it will be more informative to list the main manuscript of the specific immune cell types and the biological interpretations.

A2: Thanks for your suggestions. As the association between candidate genes and immune cells in ESCC (Supplementary Figure 5F) is not directly related to the primary focus of this article, which revolves around these genes' association with ESCC progression risk, we have decided to exclude this supplementary figure from our study.

Q3: Please describe the machine learning and prediction models in the main text (lines 234 - 236). Currently, there is only one sentence in the main manuscript. Readers may not always look into the method section. So, more details are always more helpful.

A3: Thank you for the feedback. We have revised this section and added more information regarding the description of machine learning and prediction models both in main manuscript (page 10-11, line 385-425) and the method section with the title Machine learning models construction (page 23).

Q4: For example, is the prediction model trying to predict N, L, and M states from the gene expression of a sample itself? Or is the prediction model trying to predict whether the sample will progress into L or M using only gene expression from “N” sample?

A4: Our prediction model was developed to forecast the N, L, and M states based on gene expression data obtained from a sample itself. This information has been added to main manuscript (page 10, line 387-389). The prediction model here cannot determine whether a sample will progress into L or M only using gene expression data obtained

from “N” sample. During the revision process, we reanalyzed the immunohistochemical results using an immunohistochemical scoring system method to better align our approach with clinical diagnosis. As a result, we updated figures presented in relevant sections of the manuscript, including the prediction model.

Q5: Also, why did the logistic regression and KNN perform better than other models?

A5: In our study, we sought to ascertain the algorithms that exhibited optimal efficacy in classification tasks by assessing both logloss values and multiclass AUC values. Logloss, also known as logarithmic loss, is a scalar that assesses the classifier's accuracy in predicting class probabilities for classification tasks. It calculates the deviation between predicted and actual target probabilities. Improved algorithm performance is demonstrated by lower logloss values. Multiclass area under the curve (multiclass AUC) is a metric utilized to evaluate classification model performance when handling multiple classes. It measures how well the model can distinguish between these classes. Higher multiclass AUC values indicate that the algorithm is better at distinguishing between different classes. After thorough evaluation, we identified two algorithms knn and logre that displayed superior performance than other algorithms across both metrics, characterized by a smaller average logloss value and a higher AUC value. This information has been added to both main manuscript (page 1011, line 390-414) and the method section with the title Machine learning models construction (page 23)

REVIEWERS' COMMENTS

Reviewer #1 (Remarks to the Author):

All previous question are properly answered and the manuscript is changes accordingly.

The only remaining point is that candidate indicators is just another term for biomarker while I think the authors should focus on the biology of malignant transition and not so much on biomarkers or indicators.

Reviewer #2 (Remarks to the Author):

The auhors have adequately addressed my concerns.

Reviewer #3 (Remarks to the Author):

The authors have comprehensively addressed my concerns in the previous round. Although the TAGLN2 rescue experiment result is suboptimal, I can understand the difficulty of having perfect data in biology in general. I will be OK to see this study published.

REVIEWERS' COMMENTS

Reviewer #1 (Remarks to the Author):

All previous question are properly answered and the manuscript is changes accordingly.

The only remaining point is that candidate indicators is just another term for biomarker while I think the authors should focus on the biology of malignant transition and not so much on biomarkers or indicators.

Answer: Thank you for your suggestions. Based on your suggestion and the title proposed by the editors, we have made a modification to the title of our manuscript. The revised title is “Spatial transcriptomics analysis of esophageal squamous precancerous lesions and their progression to esophageal cancer”. We focused on the

biology of malignant transition rather than solely investigating biomarkers or indicators.

Reviewer #2 (Remarks to the Author):

The authors have adequately addressed my concerns.

Answer: We are pleased to hear that the revised version has met with your approval.

Thank you for your feedback.

Reviewer #3 (Remarks to the Author):

The authors have comprehensively addressed my concerns in the previous round.

Although the TAGLN2 rescue experiment result is suboptimal, I can understand the difficulty of having perfect data in biology in general. I will be OK to see this study published.

Answer: We appreciate your positive feedback regarding the revised version.

Thank you for your valuable input.